# Natural sounds can be reconstructed from human neuroimaging data using deep neural network representation

Jong-Yun Park[1,2]¤*, Mitsuaki Tsukamoto[2], Misato Tanaka[1,2], Yukiyasu Kamitani[1,2]*

1 Department of Intelligence Science and Technology, Graduate School of Informatics, Kyoto University, Kyoto, Japan, 2 Department of Neuroinformatics, ATR Computational Neuroscience Laboratories, Kyoto, Japan

¤ Current address: Department of Psychiatry and Behavioral Sciences, Graduate School of Medical and Dental Sciences, Institute of Science Tokyo, Tokyo, Japan
* park_jy.psyc@tmd.ac.jp (J-YP); kamitani@i.kyoto-u.ac.jp (YK)

## Abstract

Reconstruction of perceptual experiences from brain activity offers a unique window into how population neural responses represent sensory information. Although decoding visual content from functional MRI (fMRI) has seen significant success, reconstructing arbitrary sounds remains challenging due to the fine temporal structure of auditory signals and the coarse temporal resolution of fMRI. Drawing on the hierarchical auditory features of deep neural networks (DNNs) with progressively larger time windows and their neural activity correspondence, we introduce a method for sound reconstruction that integrates brain decoding of DNN features and an audio-generative model. DNN features decoded from auditory cortical activity outperformed spectrotemporal and modulation-based features, enabling perceptually plausible reconstructions across diverse sound categories. Behavioral evaluations and objective measures confirmed that these reconstructions preserved short-term spectral and perceptual properties, capturing the characteristic timbre of speech, animal calls, and musical instruments, while the reconstructed sounds did not reproduce longer temporal sequences with fidelity. Leave-category-out analyses indicated that the method generalizes across sound categories. Reconstructions at higher DNN layers and from early auditory regions revealed distinct contributions to decoding performance. Applying the model to a selective auditory attention ("cocktail party") task further showed that reconstructions reflected the attended sound more strongly than the unattended one in some of the subjects. Despite its inability to reconstruct exact temporal sequences, which may reflect the limited temporal resolution of fMRI, our framework demonstrates the feasibility of mapping brain activity to auditory experiences—a step toward more comprehensive understanding and reconstruction of internal auditory representations.

**Data availability statement:** The VGGSound dataset is available at https://www.robots.ox.ac.uk/~vgg/data/vggsound/. The raw fMRI data are also available on OpenNeuro at https://openneuro.org/datasets/ds006319 (doi: https://doi.org/10.18112/openneuro.ds006319). The preprocessed fMRI data and pre-trained model are available at https://figshare.com/articles/dataset/23633751 (doi: https://doi.org/10.6084/m9.figshare.23633751.v10). The Python codes used in this paper are available from our repository: https://github.com/KamitaniLab/SoundReconstruction/releases/tag/1.2 (doi: https://doi.org/10.5281/zenodo.15066806).

**Funding:** This research was supported by the KAKENHI grants from the Japan Society for the Promotion of Science (JSPS; https://www.jsps.go.jp), with grant numbers JP25H00450, JP20H05705 and JP20H05954 assigned to YK. Additional financial support was provided by the New Energy and Industrial Technology Development Organization (NEDO; https://www.nedo.go.jp) under the grant number JPNP20006 to YK. Furthermore, YK also received backing through JST CREST (https://www.jst.go.jp/kisoken/crest/) with grant number JPMJCR22P3. Despite this financial aid, the funders had no influence on the study design, data collection and analysis, decision to publish, or preparation of the manuscript

**Competing interests:** The authors have declared that no competing interests exist.

**Abbreviations:** AC, auditory cortex; CI, confidence interval; CNN, convolutional neural network; DNNs, deep neural networks; EEG, electroencephalography; fMRI, functional MRI; HCP, Human Connectome Project; HNR, harmonic-to-noise ratio; MEG, magnetoencephalography; ROIs, regions of interests; SC, spectral centroid; SNR, signal-to-noise ratio; VQVAE, Vector Quantized Variational Autoencoder.

## Introduction

Reconstructing perceptual content from brain activity allows us to explore how a neural population represents a coherent experience, not just isolated sensory features, revealing characteristics of stimulus coding, sensory processing, and top-down influences [1,2]. Recent advances in machine learning-based neuroimaging analysis have made it possible to decode visual content from functional MRI (fMRI) responses, allowing researchers to infer not only what individuals see [1] but also imagine [3,4] and dream [5]. Building upon these decoding approaches, recent studies have further succeeded in reconstructing visual images from brain activity [6,7]. However, while these approaches excel in the visual domain, their direct transfer to the auditory domain faces distinct challenges stemming from the richly varying temporal structure of sound and the coarse temporal sampling of fMRI.

Although various decoding strategies have been applied to the auditory system [8,9], reconstructing arbitrary sounds is more demanding than decoding images. This difficulty reflects the diversity and rapid temporal changes of natural sounds, which fMRI cannot fully capture. Consequently, many existing fMRI-based studies have focused on classification of simpler acoustic [10] or decoding of linguistic elements rather than full-scale sound reconstruction [11].

Other neuroimaging modalities, such as electroencephalography (EEG) and magnetoencephalography (MEG), do offer high temporal resolution but reduced spatial precision. While these techniques have enabled decoding of simpler or predefined speech segments [11–13], these methods remain constrained, and the lack of robust generative approaches underscores the current difficulty of capturing the full spectrotemporal complexity of natural sounds. Invasive recordings, such as intracranial electrodes, can capture both high temporal and spatial resolutions to decode overt and imagined speech [14,15] and reconstruct speech from the nonprimary auditory cortex (AC) [16]. Building on this foundation, recent studies have leveraged advanced machine learning models to reconstruct speech from the AC [17,18] and motor cortex [19,20], as well as songs [21]. Despite these advances, the broader applicability of invasive approaches remains limited due to their clinical constraints and lack of scalability.

To address these constraints, recent research has shifted toward reconstructing unconstrained auditory signals from fMRI data using feature representations, thereby eliminating the need for precise temporal alignment between neural recordings and auditory stimuli. Santoro and colleagues [22] demonstrated that the spatial patterns in fMRI could compensate for its limited temporal resolution, enabling the prediction of detailed temporal aspects of auditory features. They devised a computational model that incorporates many multivariate decoders to estimate spectral-temporal modulation features derived from 7 T fMRI activation patterns. Notably, these trained decoders successfully decoded fine modulation fluctuations within the broad temporal intervals of fMRI scans. However, despite these advancements, the reconstructed sounds from the decoded features lacked complex spectrotemporal patterns. This limitation resulted in temporally smoothed reconstructions that posed significant recognition challenges for human listeners.

An emerging line of research in sensory neuroscience has focused on leveraging representations learned by deep neural networks (DNNs) to explore parallels between artificial models and the brain's auditory processing. The hierarchical organization of DNNs, resembling that of biological sensory systems, has contributed to the rise of NeuroAI—a field that integrates artificial and biological neural networks to investigate brain function. Encoding studies have shown that DNN features can predict neural responses, providing insight into hierarchical auditory processing. Kell and colleagues [23] developed a DNN model reflecting the hierarchical structure of the human auditory pathway, showing that early AC responses aligned with lower DNN layers, while non-primary regions corresponded to higher, task-specific layers. However, such parallels are not universal, as encoding performance depends on model architecture and training objectives [24]. Giordano and colleagues [25] further demonstrated that intermediate DNN layers outperformed acoustic and semantic models in predicting neural and behavioral responses to natural sounds. These encoding studies lay the groundwork for decoding approaches, which reconstruct stimuli from brain activity and provide complementary perspectives on auditory representation. Investigating reconstruction fidelity across auditory regions further clarifies their distinct contributions to sound perception and neural coding.

Building on these insights, we hypothesize that hierarchical DNN features—derived from models designed to mimic the organization of the human auditory pathway—can support perceptually reliable sound reconstructions, even without precise temporal alignment between neural responses and auditory stimuli. To test this, we use a convolutional neural network (CNN) model optimized for sound recognition to extract the DNN features that encode acoustic information across multiple temporal and spectral scales (Fig 1A). These high-level features, which integrate information over broad receptive fields, allow us to reconstruct sounds from fMRI responses without relying on exact timing correspondence between neural activity and auditory input. To evaluate whether these features are explicitly represented in fMRI responses, we employed a linear decoding approach. This strategy tests whether information can be directly extracted from localized neural activity using linear transformations alone, providing a conservative estimate of representational content. A critical component of the sound reconstruction involves converting the decoded DNN features into audio. To achieve this, we train an audio generator composed of interconnected models to convert decoded DNN features into spectrograms and transform the spectrograms into audio waveforms. Once trained, we apply the complete pipeline, where fMRI responses are decoded into DNN features and processed by the audio generator, thereby completing the reconstruction from brain activity to sound (Fig 1B).

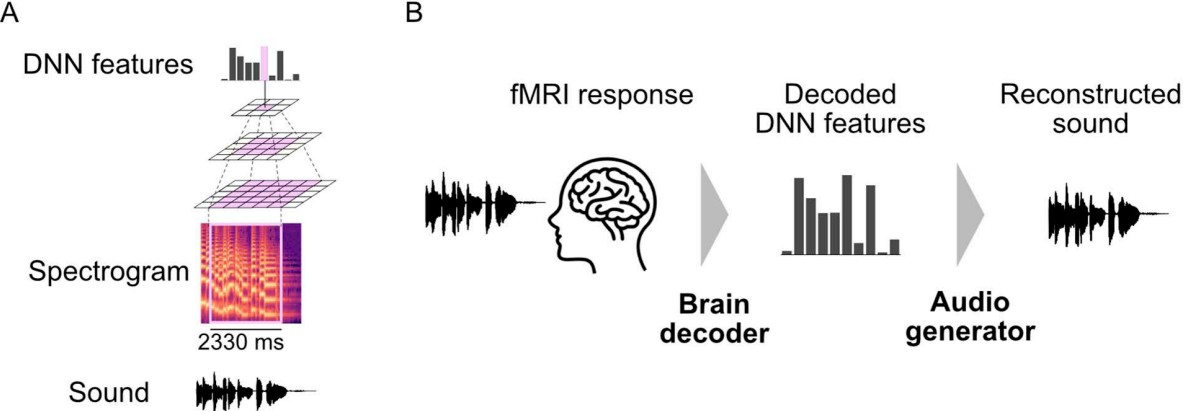

**Fig 1. Schematic overview of the proposed sound reconstruction pipeline. (A)** DNN feature extraction from sound. A deep neural network (DNN) extracts auditory features at multiple levels of complexity using a hierarchical framework. **(B)** Sound reconstruction. The reconstruction pipeline starts by decoding DNN features from fMRI responses using trained brain decoders. The audio generator then transforms these decoded features into the reconstructed sound.

Our analyses provide several key findings that collectively validate the proposed reconstruction framework. First, we demonstrate that DNN features can be decoded from fMRI responses with greater accuracy than traditional spectrotemporal features. Using these decoded features, an audio generator reconstructs perceptually plausible sounds, as confirmed by both human ratings and quantitative feature analyses. These results indicate that high-level DNN features with broad receptive fields enable sound reconstruction from fMRI without requiring precise temporal alignment. However, while the reconstructed sounds from fMRI responses preserve perceptual characteristics, they are limited in capturing long temporal sequences. To examine the extent of temporal preservation, we analyze reconstructions using temporally disrupted stimuli and found that short-term spectral patterns are retained, whereas longer temporal structures are not fully reconstructed. In addition, leave-category-out analyses demonstrate that the brain decoder generalizes across sound categories, maintaining robust performance even for previously unseen categories. Further comparisons across auditory regions and DNN layers highlight their respective contributions to reconstruction fidelity. Finally, as part of an experimental extension, we apply our model to a "cocktail party" scenario to examine its capability to reconstruct a focused sound amidst competing auditory stimuli.

## Results

We conducted fMRI experiments on five healthy subjects (S1–S5) to investigate neural responses to natural sound stimuli. Data from S1 were used for exploratory analysis to establish the reconstruction pipeline and optimize parameters. The remaining four subjects then independently validated the results using the finalized analysis settings determined from S1. Each subject underwent whole-brain fMRI scans (TR = 2 s) while listening to 8-s excerpts of natural sounds (10-s for S1). For the training dataset, we presented 1,200 natural sound stimuli spanning diverse and mixed sound categories, each repeated four times (Fig 2A). For the test dataset, we selected 50 natural sounds categorized into human speech, animal sounds, musical instruments, and environmental sounds based on prior research [26] (Fig 2B and 2C). Each test stimulus exclusively represented a single category. The sound stimuli used for fMRI recordings were distinct from those in the DNNs and generator model training dataset (see "Materials and methods: Stimuli").

Data samples for machine learning analyses were created by pairing sound segments and the fMRI responses in the auditory cortical areas. To augment the number of data samples, each 8-s sound stimulus was divided into three overlapping 4-s windows (Fig 2D). Each 4-s window was matched to the average of three fMRI volumes recorded 2–8 s after stimulus onset, thereby discarding any internal temporal sequence within the window. This approach increased the number of training samples to 14,400 (1,200 stimuli × 4 repetitions × 3 samples = 14,400). For testing, each stimulus was presented eight times, with analyses performed on either single-trial data or 8-trial averages to enhance the signal-to-noise ratio (SNR). The primary results shown here use the 8-trial-averaged samples (50 stimuli × 3 samples = 150) while single-trial results are shown in additional analyses. Although using overlapping time windows improves decoding performance by augmenting training data size, it also introduces dependence among the three samples within each 8-s stimulus block. To ensure statistical independence, prediction accuracies from each window were first obtained and then averaged, yielding 50 independent test data points (see "Materials and methods: fMRI preprocessing and data sample construction"). Neural activity was examined across 13 auditory cortical regions, as defined by the Human Connectome Project (HCP) [27]. To delineate a broader AC, we combined responses from all 13 regions of interest (ROIs) (Fig 2E).

### Brain decoding of auditory features

We first conducted feature decoding analysis using DNN features extracted from auditory stimuli using VGGish-ish [28], a CNN optimized for sound recognition. These features, which served as decoding targets, encapsulated acoustic information across multiple temporal and spectral scales, ranging from fine spectrotemporal details to broader acoustic structures. Based on preliminary results, we selected the Conv5 layer—the highest convolutional layer—as the primary decoding target, as it encodes high-level auditory representations. For each subject, we trained a brain decoder to predict individual

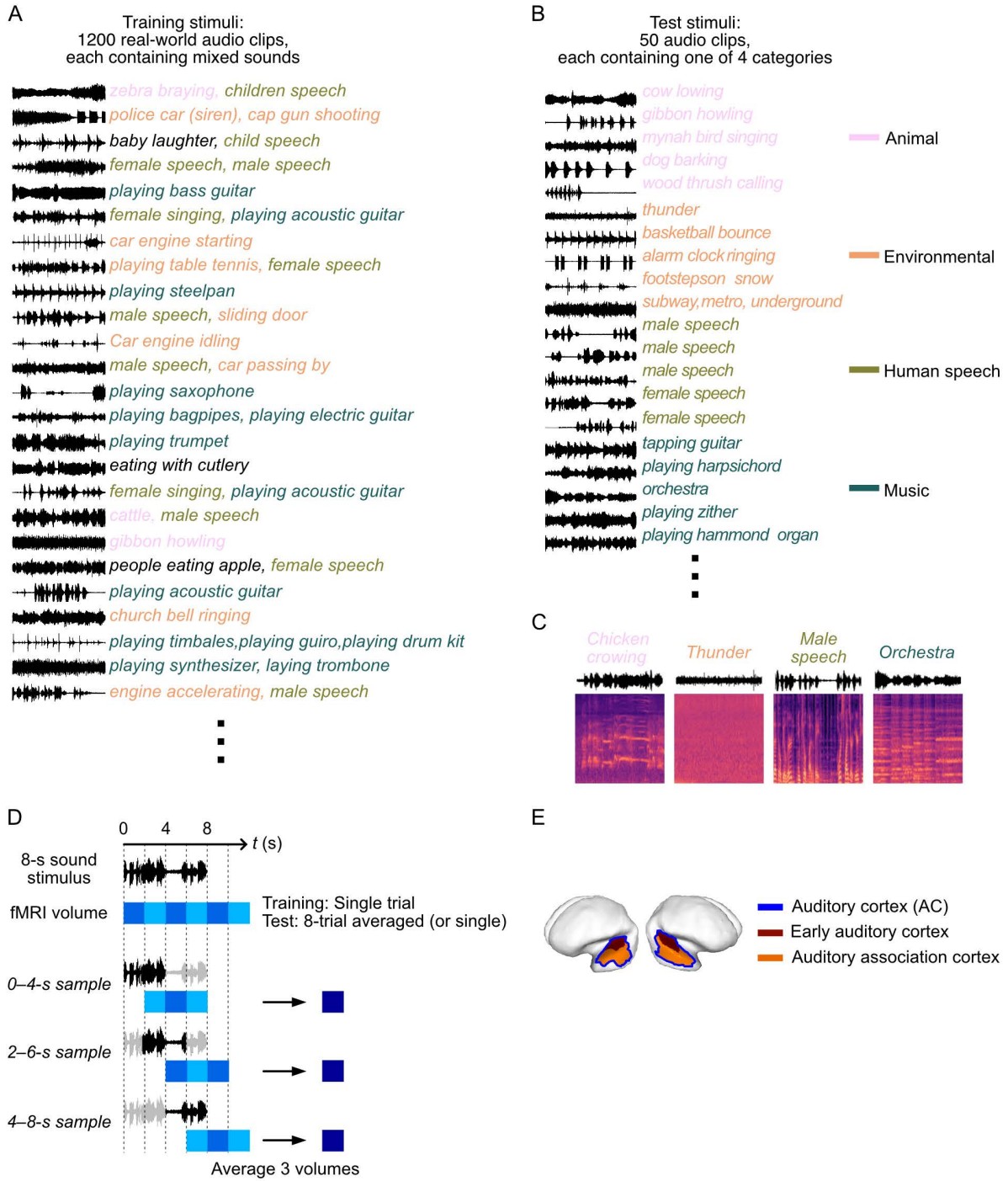

**Fig 2. Sound stimuli and brain data. (A)** Training dataset. Waveform examples and category labels are displayed, with each category—human speech, animal sounds, musical instruments, and environmental sounds—represented in distinct colors. **(B)** Test dataset. Examples selected from each of the four categories are shown, with corresponding waveforms and labels. **(C)** Examples of spectrograms from the test dataset. One example from each category is displayed. **(D)** Data samples for machine learning analyses. Each 8-s sound stimulus is divided into three overlapping 4-s windows, and corresponding fMRI responses (3 volumes) are averaged within each window to create data samples. Single-trial fMRI volumes are used for training data, while test data utilize either single-trial volumes or volumes averaged across eight repetitions. **(E)** Definition of auditory cortex (AC). The AC, outlined by blue lines, is delineated as a combination of two regions: the early auditory cortex, shown in brown, and the auditory association cortex, shown in orange.

DNN features from voxel patterns of fMRI responses using L2-regularized linear regression. The trained model was applied to decode the DNN features of test samples, and the decoding performance was evaluated by calculating profile correlations and assessing the ability of decoded features to identify the perceived sounds within the test set (Fig 3A). To assess whether the network's hierarchical structure alone could explain the observed decoding performance, we compared results using DNN features from an untrained model. Furthermore, we compared the results with those of traditional auditory features, such as spectrogram and spectrotemporal modulation features, to assess their relative strengths and limitations.

We first evaluated decoding performance by calculating the Pearson correlation between the true and decoded feature values for each unit of DNN features across the test samples (profile correlation; Fig 3B). To assess overall performance, the correlation coefficients were averaged across units within each feature type or layer. As shown in Fig 3C, profile correlations computed from the AC are consistently positive across all subjects and auditory features, indicating effective decoding. We also found that the trained DNN consistently exhibited higher correlations across subjects compared to the untrained model, highlighting the advantages of task-specific training. To validate that these correlations reflect meaningful relationships rather than chance-level similarity, baseline profile correlations were also computed between randomly selected pairs of natural stimuli. The resulting baseline values were close to zero, confirming that the observed profile correlations are not attributable to statistical bias. However, because feature dimensionality and distribution vary across feature types, direct numerical comparisons of profile correlation values across feature types should be interpreted with caution.

To compare the utility of different types of decoded auditory features, we conducted an identification analysis to determine whether the decoded features could reliably identify the perceived stimuli within the test set. This analysis was performed pairwise, comparing each decoded feature vector to two candidate vectors: one representing the true sound

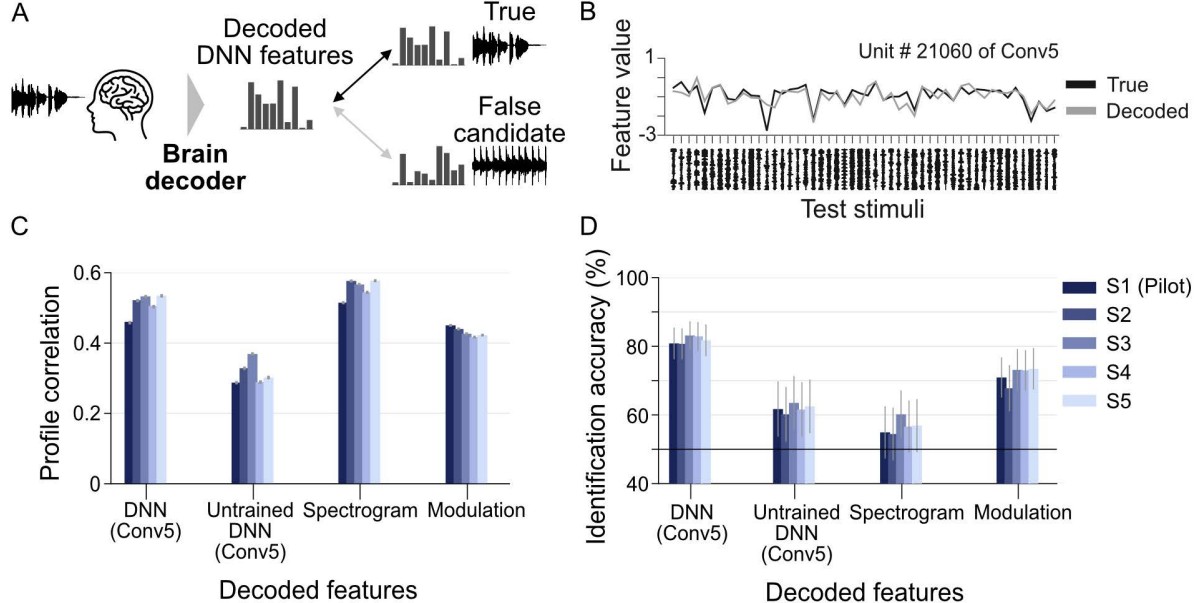

Fig 3. Feature decoding analysis. (A) DNN feature decoding and evaluation. Decoders are trained to predict DNN features from voxel patterns of fMRI responses. Decoding performance is assessed by evaluating the ability of the decoded features to identify perceived sounds from the test set. (B) Example of true and decoded features for a DNN feature unit. The graph displays the true and decoded values for a single DNN feature unit across 50 test stimuli. This unit (#21060) was from the Conv5 layer of the VGGish-ish model (ROI: AC). (C) Profile correlation of decoded auditory features. Each bar represents the mean profile correlation, with distinct colors indicating different subjects. Error bars denote the 95% confidence interval (CI). (D) Identification accuracy for decoded auditory features. Each bar represents the mean identification accuracy across 50 test stimuli, with error bars denoting the 95% CI. Although Pearson correlation was used as the primary evaluation metric, similar results were confirmed when applying Spearman correlation as an alternative measure for both profile correlation and identification accuracy. The data underlying this figure are provided in S1 and S2 Data.

and the other a lure candidate systematically selected one by one from the remaining 49 test stimuli. Decoded features were compared with the features from the true sound and those from a lure candidate based on Pearson correlation. If the correlation between decoded features and the features of the true sound was higher than that of a lure candidate, the pair was considered correctly identified. Each decoded feature was evaluated using 49 identification pairs, ensuring that each lure candidate was used once. Identification accuracy was defined as the proportion of correctly identified pairs.

Decoded DNN features from the AC consistently achieved accuracies exceeding 80% across subjects (for S1–S5, 80.8% with 95% CI [76.4, 85.3], 80.7 [76.4, 85.1], 83.1 [79.1, 87.1], 82.9 [78.7, 87.0], and 81.7 [77.3, 86.2], respectively; Fig 3D). While accuracies for decoded DNN features derived from an untrained model and decoded spectrograms were slightly above chance, those derived from modulation features reached 70% (for S1–S5, 70.9% with 95% CI [65.2, 76.5], 67.8 [61.2, 74.3], 73.1 [67.2, 79.0], 73.0 [67.3, 78.7], and 73.4 [67.5, 79.3], respectively). These results demonstrate the robustness of DNN features and their superior predictive capability in decoding auditory features, consistent with earlier encoding analyses [24].

For spectrogram features, we observed a discrepancy between profile correlation and identification accuracy: despite high profile correlations, identification accuracy remained low. This suggests that while decoded spectrograms captured common pixel variations, they failed to encode the distinct stimulus features necessary for accurate identification. This aligns with previous studies using direct regression to decode spectrograms from neuroimaging data, where decoded spectrograms exhibited a non-specific broadband component, resulting in overly smoothed patterns lacking fine-grained detail for reliable identification [29].

## Sound reconstruction

We developed a multi-stage audio generator to reconstruct sounds from the decoded features (Fig 4A). The system integrates three key components: an audio transformer, a codebook decoder, and a vocoder. Central to this architecture is a codebook system that creates efficient, compressed representations of spectrograms (S1 Fig) [28]. The codebook functions as a learned dictionary where each entry corresponds to a distinct spectrotemporal pattern derived from training data. The system processes spectrograms by dividing them into patches and mapping each patch to its nearest codebook entry, effectively reducing complex spectral data to a sequence of discrete indices while preserving essential acoustic information. For implementing this approach, we utilized the SpecVQGAN model [28] to encode and reconstruct spectrogram segments through codebook indices.

In this framework, the audio transformer bridges the gap between the decoded DNN features and codebook representations, employing autoregressive generation [30,31] to sequentially predict codebook indices across both temporal and spectral dimensions. These indices are then passed to the codebook decoder, which reconstructs the spectrogram from the discrete representation. The final component, a spectrogram vocoder, converts the reconstructed spectrogram into an audible waveform, completing the fMRI-to-sound reconstruction pipeline (Fig 4B; see "Materials and methods: Reconstruction pipelines"). To verify the pipeline's effectiveness, we conducted a recovery test by reconstructing sounds from the original DNN features of the stimuli (Fig 4B).

Fig 4C presents reconstructed spectrograms generated from brain-decoded DNN features, alongside reference spectrograms recovered from the corresponding true DNN features. The reconstructions from true DNN features (second row) exhibit high fidelity in both spectral and temporal domains, with only minimal noise artifacts compared to the original stimuli. This validates both the DNN features' capacity to encode acoustic patterns and the generator's ability to accurately reconstruct sounds when given precisely decoded features.

Individual subject reconstructions (rows three to seven; audio examples available in S1 Movie) reveal how the model captures distinct acoustic signatures across various sound categories. Animal sounds (first column) maintain their characteristic spectral patterns, while speech segments (second and third columns) preserve the harmonic structures typical of human vocalization, clearly distinguishing them from other acoustic categories. However, reconstructing long sequences

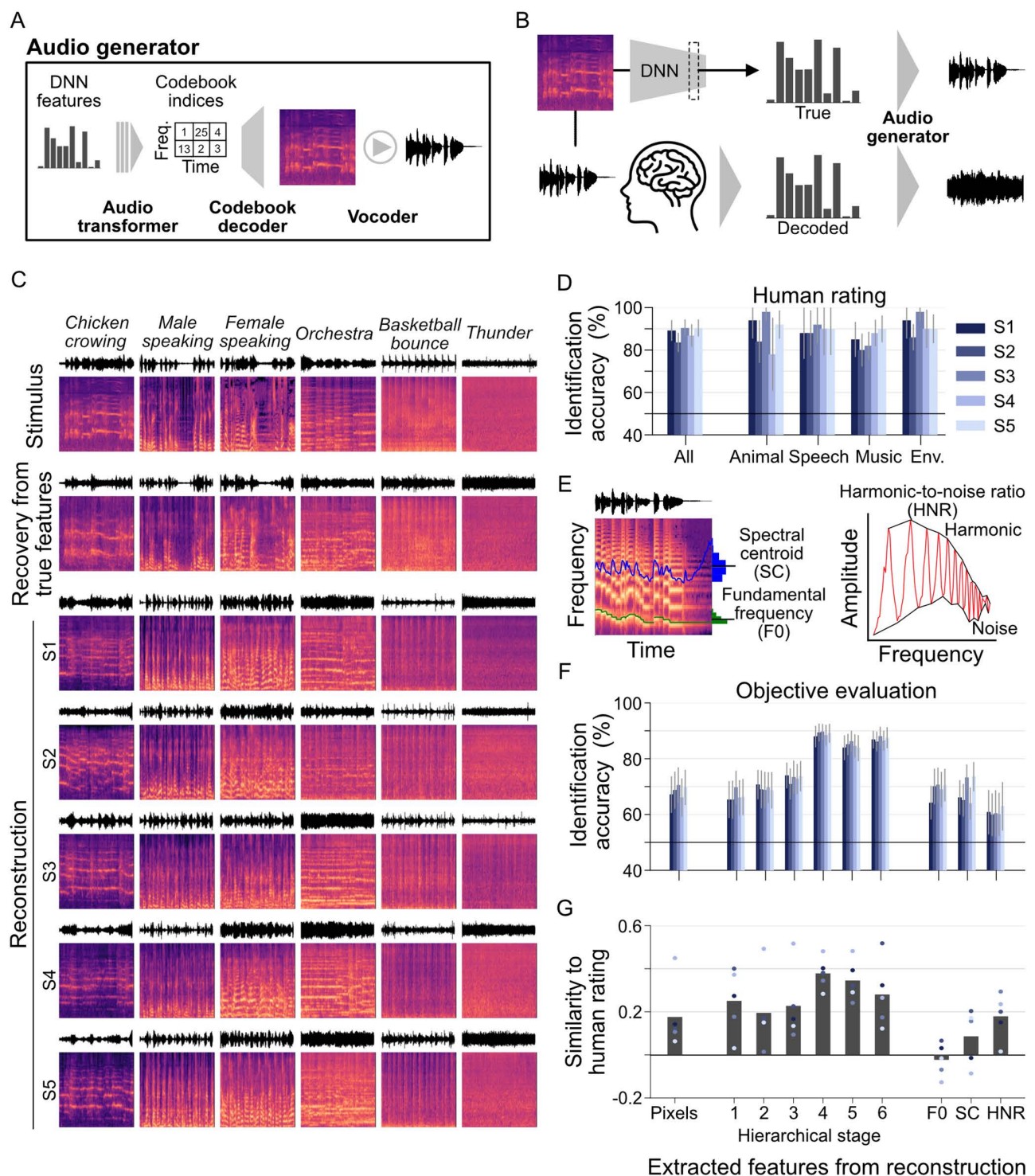

**Fig 4. Sound reconstruction. (A)** Audio generator. A multi-stage system is used to reconstruct sounds from DNN features. The audio transformer is trained to generate compact spectrogram representations as sequences of codebook indices based on decoded DNN features. The codebook decoder reconstructs the spectrogram from these indices, which are then converted into a time-domain waveform using the spectrogram vocoder. **(B)** Reconstruction pipeline. The audio generator transforms decoded DNN features from fMRI responses into sound. A recovery check is performed using the true DNN features of the stimulus. **(C)** Reconstructed spectrograms (ROI: AC, DNN layer: Conv5; for reconstructed sounds, see https://www.youtube.com/

watch?v=kNSseidxFJU). The top row shows the original spectrograms of the stimulus sounds. The second row displays the recovered spectrograms using true DNN features. The following five rows present reconstructed spectrograms derived for each subject. **(D)** Identification accuracy of reconstructed sounds based on behavioral evaluation. Bars represent the mean accuracies of pairwise identification evaluations, averaged across samples for evaluation pairs. Error bars indicate the 95% CI, calculated from 50 data points. For category-specific analyses, 20 data points were used for speech and 10 data points for the other categories. **(E)** Acoustic features. Three key acoustic features are evaluated: the Fundamental Frequency (F0), the Spectral Centroid (SC), and the harmonic-to-noise ratio (HNR). **(F)** Evaluation of reconstructed sounds based on objective feature-based measures. **(G)** Similarity of evaluation metrics with human ratings. The similarity between identification accuracy from human ratings and an objective evaluation metric is assessed using Pearson correlation across the identification accuracy of 50 reconstructed data points. Each dot represents a subject, and the bars indicate the mean across five subjects. The data underlying this figure are provided in S2 Data.

of temporal information, such as speech onset and offset timing, remains challenging. Musical excerpts (fourth column) retain their complex spectral patterns across broad frequency ranges, but the lack of precise timing details, such as rhythm, results in a smeared time span, causing multiple instrumental sounds to become texturally merged. Environmental sounds (final two columns) maintain their unique spectrotemporal characteristics, reflecting the original acoustic properties.

As demonstrated in S1 Movie, the reconstructed audio examples preserve the quality of original sounds, though lengthy sequences and specific content (such as speech and musical passages) remain challenging to recognize. Nevertheless, each reconstruction captures the distinctive acoustic qualities of its corresponding stimulus, showing consistent performance across both subjects and sound categories.

To evaluate reconstruction quality, we conducted a behavioral rating experiment using a pairwise identification task (S2 Fig; see "Materials and methods: Identification by human raters"). Human raters listened to a reconstructed sound alongside two candidate sounds: the original stimulus and a randomly selected alternative from the test set. Raters were asked to identify which candidate sound more closely matched the reconstruction. The task included comparisons both within and across stimulus categories, with reconstruction accuracy measured as the percentage of correct identifications across all candidate pairs. A group of 17 raters participated in the evaluation, assessing each reconstructed sound against its designated candidate pairs. For each subject's reconstructions, we computed the mean accuracy and confidence interval based on the 50 data points, corresponding to the number of test stimuli.

Identification accuracy was consistently higher than chance across all subjects (for S1–S5, 89.2% with 95% CI [84.6, 93.8], 83.6 [79.3, 87.9], 90.4 [86.7, 94.2], 86.8 [81.7, 91.9], and 90.4 [86.7, 94.2], respectively; Fig 4D). To further examine performance across categories, we conducted category-specific analyses in which identification scores were calculated by averaging across both within-category and across-category trials. This analysis revealed that three subjects achieved mean identification accuracies above 90% for the Animal category, all subjects scored over 80% for Speech, four subjects exceeded 90% for Music, and all subjects scored above 85% for Environment. These results demonstrate that the reconstructed sounds closely aligned with human auditory perception.

To evaluate the objective quality of reconstructed sounds, we performed a pairwise identification analysis using features extracted from the reconstructions. This analysis compares the extracted features of a reconstructed sound with those of the true spectrogram and those of a lure spectrogram. We assessed multiple feature types: raw spectrogram pixels, hierarchical features from an independent sound DNN (Melception classifier [28]), and standard acoustic metrics including fundamental frequency (F0), spectral centroid (SC), and harmonic-to-noise ratio (HNR), as shown in Fig 4E. Following the same approach used in feature decoding analysis, we used 49 identification pairs for each reconstructed sound, each consisting of the original stimulus and a lure candidate selected from the remaining test stimuli. Identification accuracy was calculated as the percentage of pairs where the reconstructed sound showed higher Pearson correlation to its original stimulus than to a lure candidate (see "Materials and methods: Evaluation of reconstructed sounds").

The evaluation results across these multiple quantitative measures are shown in Fig 4F. Analysis of spectrogram pixel values revealed above-chance identification accuracies across all subjects (for S1–S5, 67.1% with 95% CI [60.8, 73.5],

68.7 [62.3, 75.1], 70.5 [64.3, 76.7], 66.1 [59.5, 72.7], and 69.7 [63.6, 75.9], respectively), indicating preservation of raw acoustic features. Hierarchical features from an independent sound DNN demonstrated improved accuracy compared to pixel-based measures, with performance increasing at higher hierarchical stages. At the hierarchical stage 6 (category layer), all subjects achieved mean identification accuracies exceeding 85% (for S1–S5, 86.8% with 95% CI [83.9, 89.6], 86.2 [82.8, 89.5], 88.0 [84.8, 91.3], 86.5 [83.1, 89.6], and 87.5 [83.9, 91.1], respectively). The standard acoustic features showed identification accuracies comparable to spectrogram pixel analysis, with mean accuracies around 70% for F0 and SC, and 60% for HNR.

To better understand the relationship between human perception and objective feature-based measures, we analyzed correlations between identification accuracies derived from human judgments and those from objective feature comparisons across all reconstructions (Fig 4G). The analysis revealed that intermediate and higher hierarchical stages showed higher correlations with human ratings. Notably, the intermediate hierarchical stage 4 demonstrated the strongest correlation with human judgments.

Additionally, we compared the results with those of different model pipelines. The conventional pipelines, which converted brain activity into spectrograms through either direct pixel prediction or modulation features, produced smoothed patterns that poorly resembled the original spectrogram (S3 Fig and S2 Movie). Replacing only DNN features with spectrogram and modulation features while keeping other parts of our current pipeline fixed still resulted in poor performance (S4 Fig and S3 Movie). An ablation analysis, excluding either DNN features or the audio generator component from our full pipeline demonstrated that both components are necessary for optimal performance (S5 Fig and S4 Movie).

We also conducted within-category identification analysis, where candidate pairs were selected from the same stimulus category (S6 Fig). Human rating results indicate that within-category identification was generally feasible for animal and environmental sounds, but less consistent for speech and music, suggesting variability in reconstruction quality across subjects. A similar pattern was observed in the objective evaluation: reconstructions of animal and environmental sounds successfully distinguished true stimuli from lure candidates, while those of speech and music showed lower identification accuracy, reflecting a lack of fine acoustic detail necessary for within-category discrimination.

Although the primary focus of our analysis was on trial-averaged test samples, we also evaluated reconstructions generated from single-trial fMRI data. As shown in S7A Fig and S5 Movie, single-trial reconstructions exhibited more noise than trial-averaged but still retained spectral and temporal patterns. While their performance was lower than trial-averaged reconstructions, quantitative analysis (S7B Fig) confirmed that our model remains viable for achieving reasonable sound reconstruction even with single-trial fMRI samples.

## Temporal structure of reconstructed sounds

Reconstruction results (Fig 4C and S1 Movie) reveal that while the reconstructed sounds are perceptually similar to the original stimuli, they exhibit a textured quality that limits the differentiation of fine semantic and temporal structures. To investigate the temporal features preserved in the reconstructed sounds, we performed identification analyses using stimuli with disrupted temporal information.

First, we examined whether the reconstructed sounds retained only perceptually similar temporal homogeneity by comparing them to sound textures derived from the time-averaged statistics of the original stimuli (Fig 5A; Fig 5B provides an example of textured stimuli). In this identification task, the extracted features of a reconstructed sound were compared with those of two candidates: the textured true sound and an original lure candidate systematically selected from the remaining 49 test stimuli. A total of 49 evaluations were performed for each reconstructed sound.

The results show that reconstructed sounds performed worse in pixel-level and lower hierarchical stages compared to the original stimuli when evaluated against textured sounds (Fig 5C; see S8 Fig for subject-specific results). However, at the hierarchical stage 4, which aligns closely with human ratings, performance remained largely unaffected. In contrast, performance at the highest hierarchical stage 6 (category layer) declined, likely due to the difficulty of distinguishing

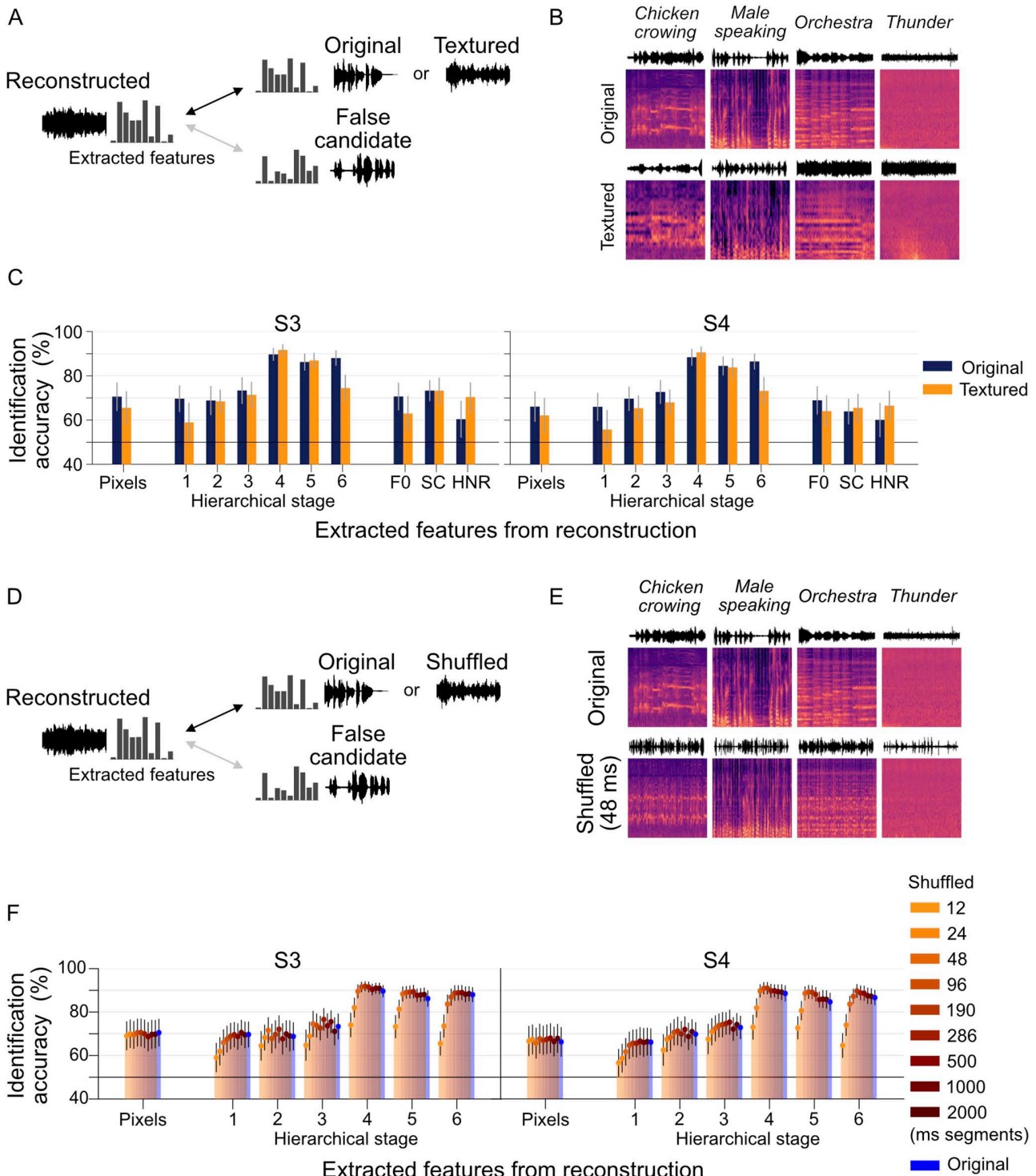

Fig 5. Evaluation using temporally perturbed stimuli. (A–C) Analysis using textured stimuli. (A) Identification analysis is conducted between the true or textured stimuli and lure candidates using extracted features. (B) The first row shows the original spectrograms of the stimulus sounds, while the second row illustrates the spectrograms of textured stimuli. (C) Each panel displays the identification accuracy for individual subjects (*e.g.*, S3, S4). The dark blue bars represent the identification accuracy for reconstructed sounds using the original true stimuli, while the orange bars indicate accuracy for reconstructed sounds using the textured true stimuli. Error bars denote the 95% CI, calculated from 50 data points. (D–F) Analysis using temporally

shuffled stimuli. (D) Identification analysis is performed between the true or shuffled true stimuli and lure candidates using extracted features. (E) The first row depicts the original spectrograms of the stimulus sounds, and the second row presents examples of temporally shuffled stimuli. Here, spectrograms are divided into equal-sized time windows (*e.g.*, 48 ms), and the segments are randomly shuffled to introduce temporal perturbations. (F) Each panel shows the identification accuracy for individual subjects. The bars represent the mean identification accuracy for various segment sizes, with different colors indicating specific segment sizes. The data underlying this figure are provided in S2 Data.

sounds with similar temporal homogeneity within the same category. These findings suggest that reconstructed sounds retain temporal information beyond simple time-averaged statistics, as evidenced by reduced performance at lower hierarchical stages and stable performance at higher hierarchical stages.

To further investigate the preservation of temporal structure in reconstructed sounds, we performed an identification analysis using perturbed versions of the original stimuli. We divided the spectrograms of the original sounds into equal-sized time windows and randomly shuffled the segments to introduce temporal disruptions (Fig 5D and 5E). To investigate the effect of temporal resolution, we repeated the analysis using several different window sizes. This analysis assessed whether reconstructed sounds could correctly identify the temporally shuffled true sound when compared against an unshuffled lure candidate, thereby evaluating the extent to which the reconstructed sounds retain the temporal structure of the original sounds. The identification task involves comparing the extracted features of a reconstructed sound with those of the temporally shuffled spectrograms of the true sound and those of an original lure candidate selected from the remaining test sounds. A total of 49 evaluations were performed for each reconstructed sound across all time window sizes, providing a comprehensive evaluation of temporal feature preservation.

As shown in Fig 5F, identification performance declined as the time window size decreased, indicating reduced sensitivity to shorter temporal disruptions. For time windows larger than approximately 100 ms, the reconstructed sounds achieved identification accuracies comparable to those of the shuffled original sounds (see S9 Fig for subject-specific results). These findings suggest that the reconstructed sounds primarily retain short-term (~100 ms) temporal features encoded in the brain, while the preservation of large-scale temporal information remains limited.

### Generalization beyond trained categories

To evaluate the generalizability of our proposed model beyond its training data, we conducted a post hoc leave-category-out analysis. In this approach, the decoder was trained with one sound category systematically excluded from the training data. The decoded features were then processed through the reconstruction pipeline to assess the model's ability to reconstruct test sounds from the omitted category.

The results demonstrate the model's capability to reconstruct spectral and temporal patterns resembling the original stimuli, even for categories excluded during training (Fig 6 and S6 Movie). The model demonstrated high fidelity in reconstructing animal and environmental sounds, despite their exclusion from training. However, omitting the music category led to a decline in perceptual performance, with the reconstructed spectrograms appearing noisier and less distinct than those of other categories, while still retaining some rhythmic spectral-temporal patterns characteristic of music. Quantitative evaluations compared identification accuracies between the left-out and full training sets. Reconstructions of animal and environmental sounds from the left-out datasets achieved accuracies exceeding 70%, comparable to the full training sets. For speech and music, identification accuracies ranged from 60% to 80% across most metrics, with a slight decline in performance. These suggest that the hierarchical DNN features used in our approach capture fundamental acoustic properties that are shared across different sound categories. By encoding these shared properties, the model can generalize to novel categories without requiring explicit training on them. However, the reduced perceptual fidelity observed for the speech and music category highlights a limitation in fully capturing its finer auditory characteristics, which may require more specialized feature representations.

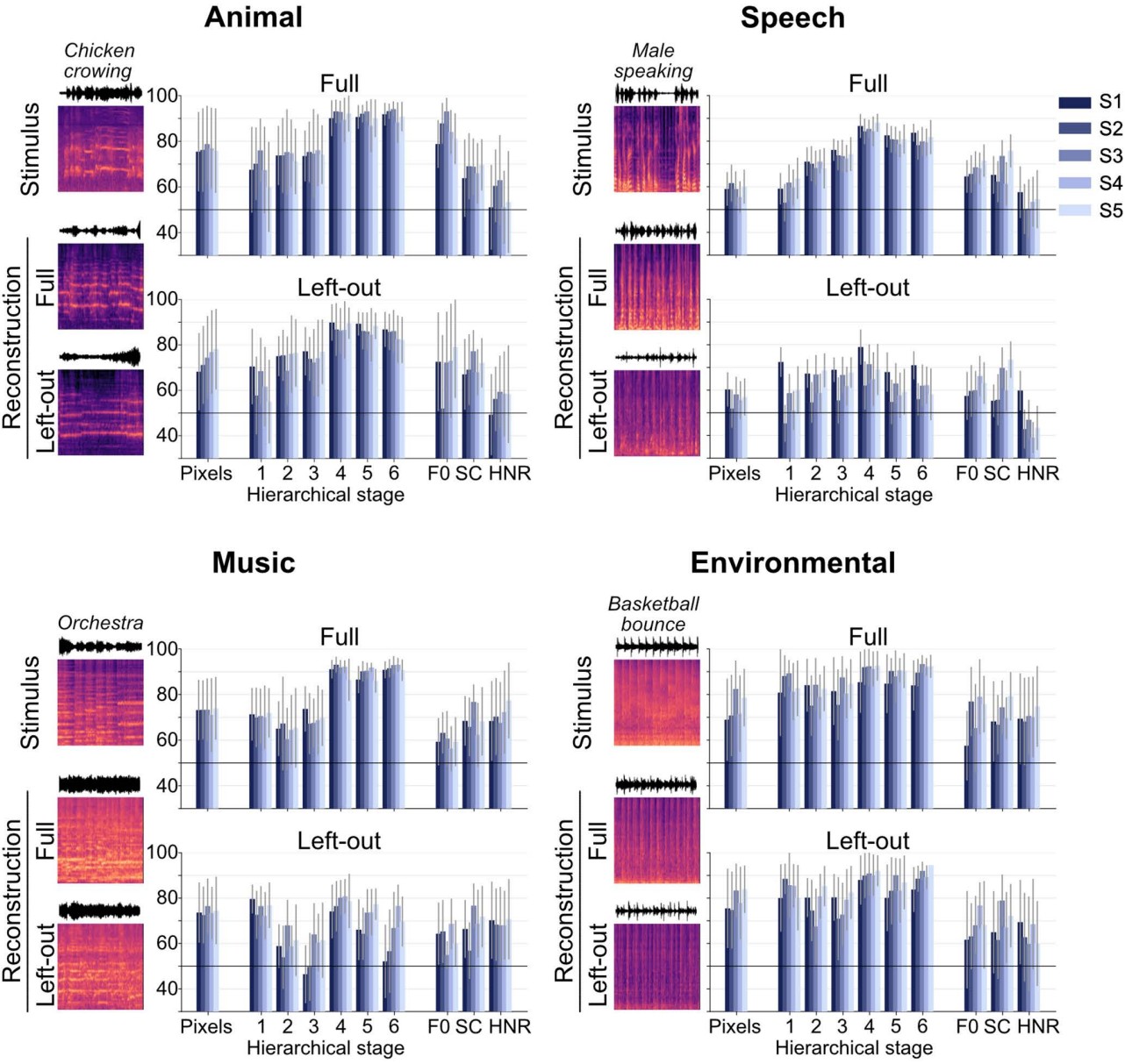

**Fig 6. Reconstructions with leave-category-out analysis.** Each panel corresponds to a sound category excluded during training. The upper row in each panel illustrates the spectrograms of the original stimuli. The middle row displays the reconstructed spectrograms generated by decoders trained on the full dataset. The bottom row depicts the reconstructed spectrograms obtained using decoders trained on a dataset where data from the test category were excluded during training (ROI: AC; DNN layer: Conv5). Audio examples of the reconstructed sounds are accessible at https://www.youtube.com/watch?v=znm6NWL1YYY. Adjacent to the spectrograms, the upper bar graphs represent the identification accuracies achieved using decoders trained on the full dataset, while the lower bar graphs show the accuracies for decoders trained on the leave-category-out dataset. Error bars represent the 95% CIs. Each color in the bar graphs corresponds to a different subject. The data underlying this figure are provided in S2 Data.

## Hierarchical auditory regions and DNN layers

In the previous sections, we validated our methods using the AC as the target ROI and Conv5 as the representative DNN layer. Here, we investigate how specific auditory regions and DNN layers influence sound reconstruction. We examined early auditory regions, including A1, LBelt, and PBelt, as well as higher auditory association areas, such as A4 and A5, which form part of the ventral auditory pathway. Additionally, we evaluated six representative DNN layers, spanning both convolutional and fully connected layers, to assess their contributions to decoding and reconstruction performance.

To analyze the contributions of individual auditory regions, we trained feature decoders separately for each ROI. Decoded features from Conv5 exhibited slightly lower identification accuracy when decoded from individual ROIs compared to the whole AC, but all exceeded 70% accuracy with minimal differences across ROIs (S10A Fig). These decoded features were then used as inputs for the audio generator. Reconstructed sounds from individual ROIs preserved short-term spectral and perceptual properties and displayed high reproducibility across regions (S11A Fig and S7 Movie). Quantitative evaluations revealed that reconstruction fidelity varied across ROIs (Fig 7A). While reconstruction from individual ROIs showed slightly reduced performance compared to the whole AC, all regions reliably supported the identification of original sounds. Higher hierarchical stages exhibited minimal differences across ROIs. However, evaluations using spectrogram and lower hierarchical stages indicated higher fidelity in early auditory regions, particularly A1 (in the hierarchical stage 1, for S1–S5, 62.4% with 95% CI [56.1, 68.7], 62.2 [55.4, 69.0], 69.7 [64.0, 75.3], 65.3 [58.6, 72.0], and 67.0 [60.4, 73.5], respectively), Fidelity decreased along the ventral pathway, with A5 showing slightly lower performance compared to A1, ranging from 56.3% with 95% CI [48.7, 64.0], 53.6 [45.9, 61.3], 54.9 [47.8, 62.0], 54.9 [47.5, 62.4], and 54.5 [47.0, 62.0] for S1–S5, in the hierarchical stage 1. These findings suggest broadly distributed neural responses across the auditory system, with some regional variation in reconstruction fidelity. Early auditory regions, such as A1, tended to exhibit slightly higher fidelity than downstream areas along the ventral auditory pathway.

We also analyzed how hierarchical representations within DNN layers influence sound reconstruction by training feature decoders to predict DNN features from each VGGish-ish layer in the AC. Decoded features showed a trend of improved performance at higher DNN layers compared to lower layers (S10B Fig). Each layer's decoded DNN features were used to generate reconstructed sounds via audio generators specifically trained for the corresponding DNN layer.

The results, summarized in S11B Fig and S8 Movie, revealed distinct patterns across layers. Reconstructions from lower layers (e.g., Conv1 and Conv2) exhibited temporally smoothed patterns that partially resembled the original spectrograms. Intermediate layers (e.g., Conv3 and Conv4) produced spectral patterns more closely aligned with the original stimuli, while higher layers, particularly Conv5, replicated the spectral patterns with greater fidelity. Even fully connected layers (e.g., FC3), despite lacking temporal dimensions, captured distinct spectral characteristics of the stimuli. We additionally evaluated FC1, the first fully connected layer after Conv5, and observed similar reconstruction trends to FC3 (S12 Fig). Overall, reconstructions from higher layers better preserved perceptual content compared to lower layers. Quantitative evaluations (Fig 7B) revealed subtle differences in spectrogram and acoustic feature metrics across layers. Lower layers achieved approximately 70% accuracy in the hierarchical stage 1, but performance declined with increasing hierarchical stages. This trend reversed for Conv4 and higher layers, where performance improved with higher hierarchical stages. These findings suggest that the proposed model effectively utilizes hierarchical DNN features, with distinct contributions from each layer to auditory perception.

To confirm whether the observed differences across layers were due to inherent limitations of the audio generator rather than the features themselves, we tested the generator's ability to recover the original sound from true DNN features at each layer (S13 Fig). Recovery from lower layers closely matched the original spectrograms, with only minor degradation at Conv4 and Conv5. These findings suggest that the variations in reconstruction quality primarily reflect the nature of the decoded features, rather than any intrinsic limitation of the audio generator.

To assess whether the hierarchical representations in reconstructed sounds arose primarily from the model's inherent architecture or from its learned features, we performed a reconstruction analysis using a VGGish-ish model initialized

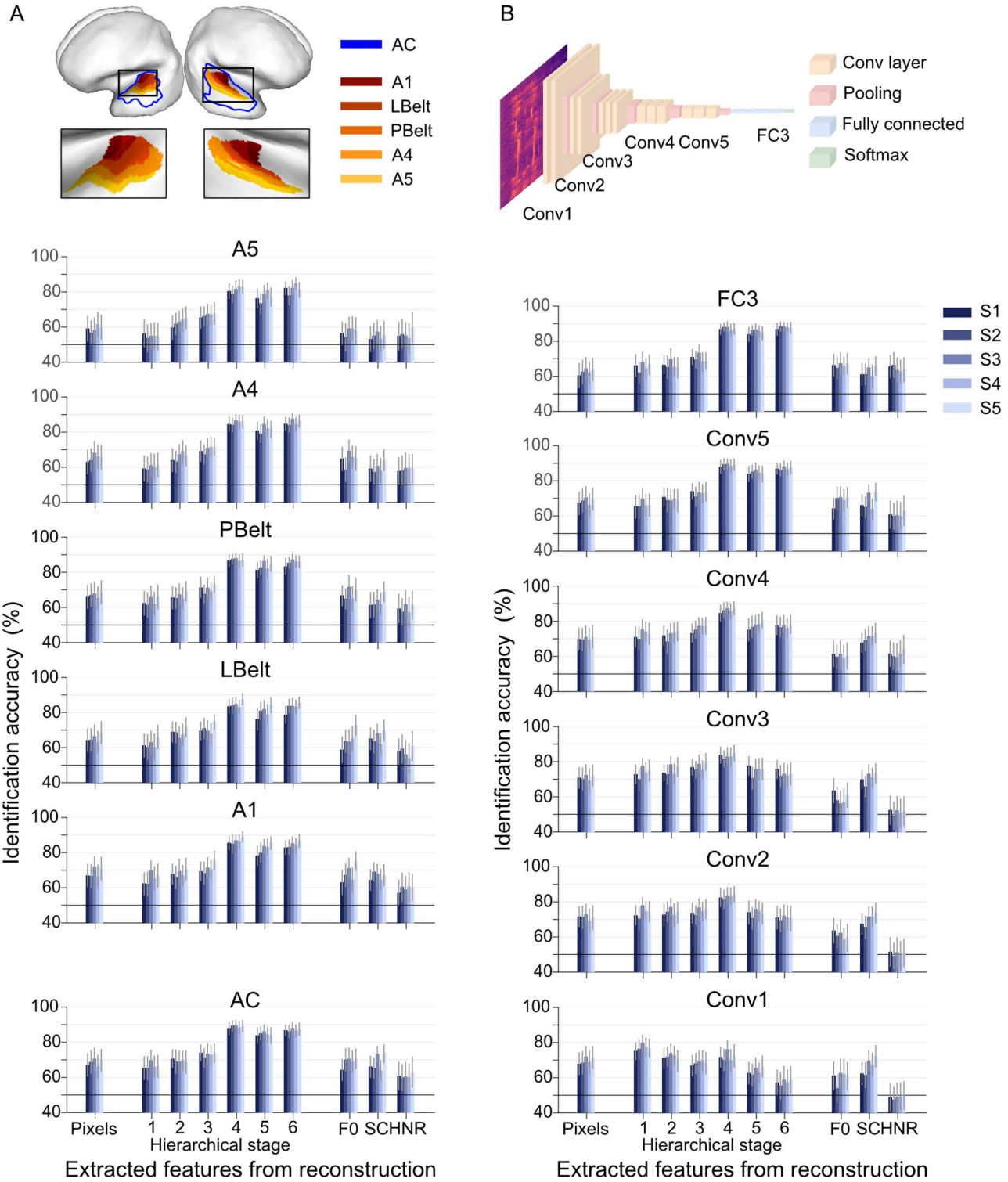

**Fig 7. Effect of hierarchical auditory regions and DNN layers. (A)** Evaluation of reconstructed sounds from individual auditory regions (DNN layer: Conv5). **(B)** Evaluation of reconstructed sounds from different DNN layers (ROI: AC). Each bar represents the mean identification accuracy calculated for each subject, with the error bar indicating the 95% CI estimated from 50 data points. The data underlying this figure are provided in S2 Data.

with random weights (S14 Fig and S9 Movie). Although some layer-wise differences in reconstruction quality could be attributed to the network's architecture, task-optimized DNN features significantly heightened these hierarchical distinctions and improved perceptual evaluations. This highlights the crucial role of learned, domain-relevant features for achieving high-quality auditory reconstructions.

Furthermore, to determine whether hierarchical evaluation metrics were shaped by task-optimized learning or merely by the model's inherent structure, we assessed reconstruction performance using an untrained evaluation model (S15 Fig). The results showed that in hierarchical stage 4—which closely aligns with human ratings—the reconstructed sound from Conv5 exhibited a significant performance decline of nearly 30% when evaluated using the untrained model. These findings underscore the critical role of task-specific optimization in capturing hierarchical sound representations, demonstrating its necessity for accurate evaluation.

## Attention

To examine whether our reconstructions captured subjective listening experiences, we recorded fMRI data in a selective auditory attention task under "cocktail party" conditions [32], which test the auditory system's capacity to segregate target sounds from the background. We paired two sounds to create 24 superimposed pairs (16 for S1) using a set of sounds (two per category, except Speech, which had four). During each trial, participants were cued visually to attend to one of the two overlapping sounds, yielding 48 attention trials in total (32 for S1). Each was repeated eight times and the corresponding functional volumes were averaged to enhance the SNR. After further data augmentation, this attention task comprised 144 test samples (24 stimuli × 2 attention conditions × 3 samples; see "Materials and methods: Stimuli, Experimental design and fMRI preprocessing and data sample construction"). For the reconstruction analysis, we used the same feature decoders trained on the single sound condition to extract DNN features from the fMRI signals recorded during the selective attention task and subsequently fed these features into our reconstruction pipeline to generate the sound under attention.

When comparing reconstructions of the same superimposed stimulus under different attention conditions, the output tended to align more closely with the attended sound while still reflecting aspects of the overall mixture (Fig 8A and S10 Movie). For instance, when attention was focused on speech, the reconstructed audio exhibited the harmonic structure typical of speech while retaining some elements of the unattended stimulus.

To assess the ability of our reconstructed sounds to distinguish between the attended and unattended stimuli, we conducted an identification analysis based on human ratings and audio features extracted from the reconstructed sounds (Fig 8B). For human ratings, raters listened to each reconstructed clip and were asked to judge whether it matched the attended or unattended stimulus more closely. The feature-based identification task involved comparing the extracted features of a reconstructed sound with those of the attended and unattended stimuli. A reconstructed sound was considered correctly identified if its correlation with the attended stimulus was higher than with the unattended stimulus. For each stimulus, the binary results from three samples in the same attention task were pooled using majority voting to generate a single binary data point, resulting in 48 binary data points per subject and evaluation condition. We calculated the mean identification accuracy from these 48 data points and compared it with the significance threshold determined by a binomial test. Unlike the previous evaluations, which allowed for multiple data points per reconstruction, this analysis generated only a single binary (correct/incorrect) result for each reconstruction. Accordingly, a binomial test against a chance-level null distribution (0.5) was used for statistical evaluation instead of computing confidence intervals.

Human ratings showed that listeners could distinguish the attended stimulus from the unattended one, achieving identification accuracies of 62.5%, 58.3%, 64.6%, 70.1%, and 58.3% for S1 through S5, respectively (Fig 8C). In contrast, a pixel-level analysis yielded accuracies below 60% (53.1%, 58.3%, 56.3%, 66.7%, and 58.3%). However, when reconstructions were evaluated at hierarchical stage 4, which had previously demonstrated strong alignment with human ratings in single-sound conditions, four subjects surpassed 65% accuracy (65.6%, 52.1%, 66.7%, 68.8%, and 66.7%). In contrast, evaluations using simple acoustic features generally failed to achieve accuracy levels above chance.

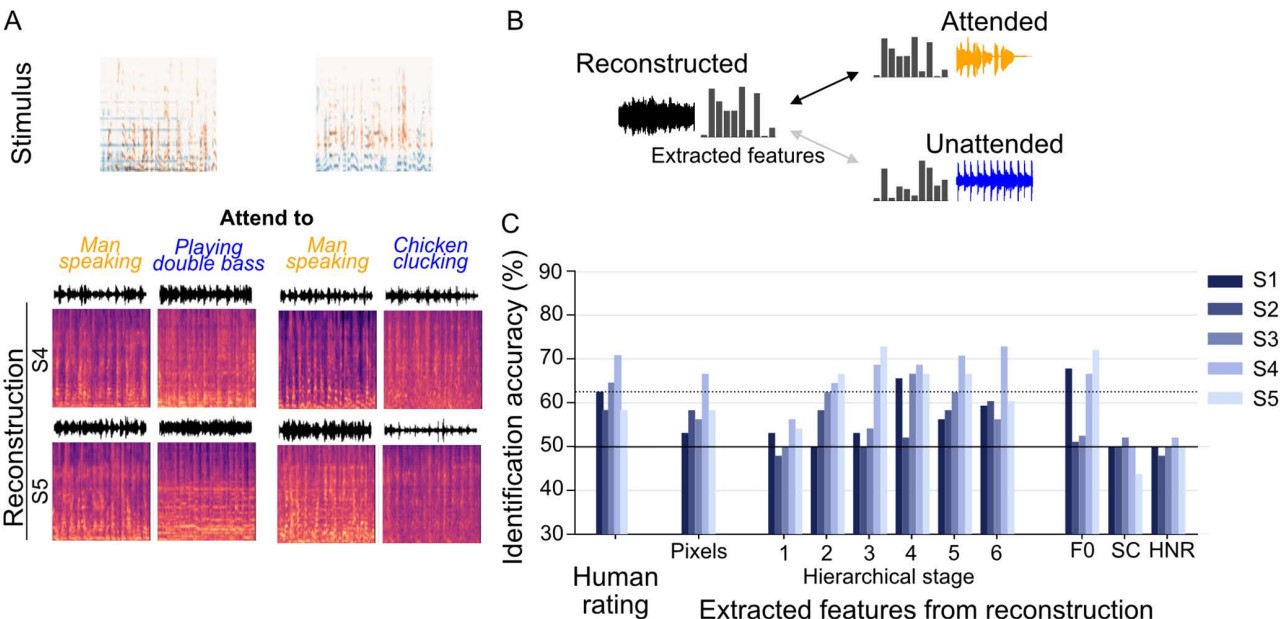

**Fig 8. Sound reconstruction with attention. (A)** Reconstructed spectrograms under selective auditory attention tasks (ROI: AC, DNN layer: Conv5; for reconstructed sounds, see https://www.youtube.com/watch?v=1ZHCoiyqPb4). The top panel displays the spectrograms of two superimposed sounds presented during the task, where subjects were instructed to focus on one specific sound. The bottom panel shows the spectrograms reconstructed by different subjects (S4, S5). **(B)** Evaluation of attentional bias. Identification analysis was conducted to evaluate the attentional bias by comparing the correlation of reconstructed features with those of the attended and unattended stimuli. **(C)** Identification accuracy of attended sound. Each bar represents the correct rate among the 48 identification trials. Since the identification of the attended vs. unattended sound was scored as a binary outcome per trial, conventional error bars are not shown. Instead, the dashed line indicates the significance level ($p < 0.05$) based on a binomial test ($N = 48$). For the pilot study S1 ($N = 32$) and cases where F0 and HNR calculations were unsuccessful, a higher significance threshold was required (not depicted here). The data underlying this figure are provided in S2 Data.

Reconstructions from individual ROIs during the selective attention task showed resemblance to the attended stimuli and broad similarity across auditory regions (S16A Fig and S11 Movie). Quantitative assessments revealed that spectrogram pixels and lower hierarchical stages did not differ in performance significantly across ROIs. However, intermediate and higher hierarchical stages displayed slightly improved performance in higher ROIs (S16B Fig). Nonetheless, notable variability emerged among individuals, underscoring the influence of individual differences on how attention shapes reconstructed sounds.

## Discussion

In this study, we introduced a framework that combines DNN feature decoding with an audio generator to reconstruct natural sounds from fMRI data. The results show that DNN features outperform traditional spectrotemporal features in decoding accuracy, as they capture both detailed spectrotemporal patterns and higher-level semantic information. Using an audio generator, we transformed decoded DNN features into audio waveforms. Reconstructed sounds based on these features captured key spectral and perceptual properties of the original stimuli, with human evaluations confirming their similarity to the originals and alignment with objective metrics. However, the reconstructed sounds struggled to replicate longer temporal sequences. While short-term temporal patterns and spectral properties were preserved, the sounds exhibited a textured quality, lacking precise temporal fidelity over seconds, as indicated by analyses using temporally disrupted stimuli. The model generalized to sound categories excluded during training, demonstrating that hierarchical DNN features effectively capture shared acoustic properties across different categories. Reconstructions from specific auditory regions

revealed distributed neural responses across the auditory system, with minor regional variations in reconstruction fidelity. In a selective auditory attention task, reconstructed sounds aligned more closely with the attended stimulus for certain subjects, reflecting neural activity associated with selective auditory attention. These findings demonstrate the potential of this approach to capture and reconstruct perceptual aspects of auditory experiences in complex listening environments.

In our analysis, DNN features from sound recognition models, which mimicked hierarchical human auditory processing, demonstrated more effective decoding performance compared to other auditory features (Fig 3C and 3D). These findings align with previous encoding studies highlighting systematic model-brain correspondence [23]. However, when decoding was performed from individual auditory ROIs, we did not observe substantial differences either in decoding performance or in the quality of reconstructed sounds (Figs 7A and S10A). This result suggests that high-level auditory features are linearly accessible from multiple cortical regions, pointing to a potentially distributed representation in the AC. This finding is in line with recent intracranial studies that proposed a distributed functional organization within the human AC, suggesting parallel information processing across the AC [33,34]. While our study utilized anatomically defined ROIs, future research may achieve a deeper understanding of the auditory hierarchy by employing voxel selection strategies based on tonotopic maps or encoding analyses.

Although no major differences in feature decoding performance were observed across individual ROIs, some variations were noted in the evaluation of acoustic features during reconstruction (Fig 7A). Specifically, the primary auditory region showed slightly better identification performance with lower hierarchical stages and acoustic features, while this performance tended to decrease in more peripheral regions. This finding aligns with previous encoding studies [25,35], which suggest that neurons in the vicinity of the primary AC are more inclined toward local integration and favor generic acoustic representations. In contrast, higher hierarchical stages showed minimal differences across auditory ROIs, with the Belt region—positioned between A1 and the auditory association cortex—showed marginally higher fidelity. These results are also in line with the view of distributed auditory processing, where different regions may contribute in distinct ways to reconstruction fidelity. Such distribution may reflect a functional balance between detailed feature representation in early regions and integrative processing in higher-order areas.

While our use of DNN features enabled effective brain decoding and sound reconstruction, precisely aligning DNN features with biological neural representation remains challenging [36]. For instance, DNNs frequently fail to generalize human-like invariances, demonstrate high sensitivity to adversarial perturbations [37], and exhibit behaviors that diverge markedly from those of biological systems [38,39]. In the vision domain, Geirhos and colleagues [40] highlighted the texture bias in ImageNet-trained DNN models, which leads these models to rely heavily on local textures rather than global shapes—a sharp contrast to human shape-centric recognition. Similarly, Conwell and colleagues [41] demonstrated that the diversity of training datasets significantly impacts model predictivity, with constrained datasets limiting generalizability and alignment with biological systems. These discrepancies underscore the need for deeper investigations into the limitations of current DNN architectures and the development of models that more closely align with the principles governing human brain function.

While our model achieved near-perfect recovery of the original sounds when using true features, the reconstructed sounds from decoded features failed to preserve intricate temporal sequences over seconds, such as intelligible speech content or musical melodies. A key reason for this limitation would be the inherently low temporal resolution of fMRI. Although the use of hierarchical DNN features with large receptive fields helps bridge this gap for short-term features (~100 ms), the fundamental temporal sampling density of fMRI may remain a bottleneck [42,43], leading to reconstructions with a perceptually "textured" quality rather than a clear temporal structure. Future work could mitigate these limitations by exploring advanced decoding methods that incorporate sequential processing. Techniques such as long short-term memory networks and recurrent neural networks have already been applied to auditory decoding tasks with other neuroimaging modalities [44–46]. Additionally, since efforts to exploit the temporal information encapsulated within fMRI signals have shown promise [47–49], another promising avenue involves applying transformers to disentangle temporal information directly in the fMRI signals, building on this study's success with transformers for feature disentanglement.

We recognize that our audio generator may bias reconstructed outputs due to both its training data and architectural design. Advanced generative AI techniques often rely on training-test overlap and can produce "hallucinated" content rather than achieving true zero-shot reconstructions [7]. Although our leave-category-out analysis showed a modest drop in performance, we still obtained robust out-of-category reconstructions. Moreover, the near-perfect recovery of original stimuli, including artificially created examples, indicates that the generator can potentially produce arbitrary sounds rather than relying solely on training-set exemplars. Nevertheless, fully generalizing beyond the training dataset in brain decoding remains an open challenge. Additionally, the transformer's method of modeling higher-level temporal sequences diverges from human auditory processes, implying that our approach depends partly on generative assumptions and may not fully mirror natural neural mechanisms.

We applied our reconstruction pipeline to a cocktail party task to examine whether the reconstructed sounds captured participants' selective listening experiences. Although several subjects' reconstructions more closely matched the attended stimulus, not all exhibited this effect. Nonetheless, our observations broadly align with previous findings that decoded auditory features—such as envelope [50,51], spectrogram [52], and trajectory [53]—tend to track the attended sound more accurately than the unattended one. Notably, most earlier studies relied on high-temporal-resolution modalities (MEG, EEG, ECoG), whereas our results suggest that even fMRI data can reveal attentional influences under certain circumstances. In line with these findings, our auditory attention analysis also showed individual variability in reconstruction fidelity. For example, subject S4, who had a significant musical background, demonstrated particularly strong reconstructions, possibly due to a heightened ability to focus on specific sounds in a multi-speaker setting. These differences underscore the impact of personal experience and attentional strategies on decoding outcomes, suggesting that future research should account for such individual factors to deepen our understanding of the psychological and neural underpinnings of auditory attention.

Our study has several limitations that should be addressed in future work. First, the small sample size of five subjects may limit the generalizability of our findings. Future studies should include larger and more diverse subject cohorts to validate the robustness of our approach. Second, the reliance on fMRI's low temporal resolution restricts our ability to capture rapid changes in neural activity, which may explain the limited preservation of long-term temporal sequences in the reconstructed sounds. Combining fMRI with high-temporal-resolution techniques like EEG or MEG could help overcome this limitation. Finally, while our model demonstrated generalization to excluded categories, the reduced fidelity for music suggests that certain sound types may require more specialized feature representations. Future work could explore hybrid models that combine DNN features with domain-specific acoustic features to improve reconstruction quality for challenging categories like music.

In conclusion, our work demonstrates the feasibility of reconstructing natural sounds from fMRI data by integrating DNN-based feature decoding with an audio-generative framework. Although the resulting audio captures critical perceptual attributes, limitations remain in preserving extended temporal sequences. Nevertheless, the consistency of human ratings and above-chance identification underscores the potential of this approach to bridge neural signals with auditory perception. By mapping subjective auditory experiences to interpretable acoustic outputs, our method holds promise for a wide range of applications—ranging from neural decoding of imagined sounds and clinical assessments of auditory hallucinations to artistic expression and foundational auditory neuroscience. Future endeavors should focus on refining temporal fidelity, improving model generalization, and further elucidating how these reconstructions reflect the richness and variability of human auditory perception.

## Materials and methods

### Experimental details

**Subjects.** Five subjects with normal hearing abilities (non-native English speakers) participated in the study, including one female subject. The average age of the subjects was 27.6 years. One subject (S1) was used for the exploratory analysis to establish the reconstruction pipeline, while the other four subjects were used to validate the results

independently. All subjects provided written informed consent before the scanning sessions. The study protocol received approval from the Ethics Committee of the Advanced Telecommunications Research Institute International (approval no: 106) and was conducted following the principles of the Declaration of Helsinki.

**Stimuli.** The fMRI experiments on natural sound perception included 1,250 audio clips, each lasting 8 s, sourced from the VGGsound test dataset [54]. The training dataset comprised 1,200 audio clips selected without considering category labels, aiming to encompass diverse natural auditory scenes. The test dataset consisted of four representative sound categories: human speech (including English), animal sounds, musical instruments, and environmental sounds. More speech stimuli were selected, totaling 20 clips, while the remaining categories each contained 10 clips. This selection process resulted in a test dataset containing 50 audio clips, each belonging to a single category. While the pilot study (S1) initially used 10-s stimuli, only the data for the first 8 s were analyzed. In addition, 81 stimuli that were difficult for human listeners to recognize were replaced with new stimuli in the subsequent experiments conducted with the remaining four subjects. All audio clips were resampled to 22,050 Hz and normalized to ensure consistent energy levels across stimuli.

During the attention session, subjects were presented with pairs of superimposed sounds drawn from four categories: male speech, female speech, animal sounds, and music. Although environmental sounds were initially included in the pilot study (S1), they were excluded from the main analysis due to difficulty in selectively attending to them when overlapped with other sounds. The pilot experiment also used both English and non-English speech stimuli. To minimize potential attentional bias arising from greater familiarity with English, stimulus pairs of English and non-English speech were excluded during the pilot study. Consequently, only non-English speech stimuli were used in the main experiment. For S1, two exemplars were randomly selected from each category. One exemplar from each category was paired with one from a different category, ensuring that each exemplar was used only once across all cross-category combinations. This procedure resulted in 16 stimulus pairs. For the remaining four subjects, two exemplars were chosen based on those demonstrating the highest reconstruction performance in the pilot study. All possible cross-category combinations were then constructed, resulting in 24 stimulus pairs. Each attention trial required subjects to focus on one of two concurrently presented sounds, generating 48 attention trials in total (32 trials for S1). The energy levels of superimposed sounds were normalized to maintain consistency.

**Experimental design.** The study included three fMRI sessions: training, test, and attention sessions. Sound stimuli were presented through fMRI-compatible headphones (Kiyohara, KAS-3000HK) at approximately 68–75 dB SPL. fMRI responses were continuously recorded throughout stimulus presentation without silent interstimulus gaps. Before scanning, subjects adjusted the sound level for comfort.

During the training session, 1,200 training stimuli were presented while whole-brain fMRI responses were recorded. Each subject participated in 12–16 scanning sessions over approximately three months, with each session consisting of 4–8 functional runs lasting up to 90 min. Each run began with a 30-s rest period, followed by 55 stimulus presentation blocks of 8 s each, including 50 unique stimuli and five randomly interspersed behavioral task blocks. For S1, stimuli lasted 10 s. A 10-s rest period followed the last stimulus, resulting in an 8-minute run duration. Training sessions were repeated four times. To maintain concentration, a one-back repetition detection task was included, requiring subjects to press a button when a stimulus was repeated. Repetition blocks (five per run) were excluded from analysis.

For the test session, 50 test stimuli were presented in a single run, repeated eight times within a single session, following the same protocol as the training session.

During the attention session, superimposed sound stimuli were presented under diotic listening conditions. The session included eight runs, each containing 48 attention trials. In each trial, a pair of sounds from different categories was presented for 8 s, and subjects focused on one of the two sounds. A total of 24 unique sound pairs were used, with each pair appearing under two attention conditions, depending on the attended sound. Visual cues indicated

the target sound by displaying both category names with a dash ("-") next to the attended category. Between trials, a behavioral task was randomly introduced four times per run, requiring subjects to identify the attended sound category from the previous trial.

**MRI acquisition.** fMRI data were acquired using a 3.0-Tesla Siemens MAGNETOM Verio scanner at the Kyoto University Institute for the Future of Human Society. An interleaved T2*-weighted gradient-echo echo planar imaging (EPI) sequence was used to obtain functional images that covered the entire brain (TR = 2,000 ms, TE = 44.8 ms, flip angle = 70 deg, FOV = 192 × 192 mm, voxel size = 2 × 2 × 2 mm, slice gap = 0 mm, number of slices = 76, multiband factor = 4). T1-weighted magnetization-prepared rapid acquisition gradient-echo (MP-RAGE) fine-structural images of the entire head were also obtained (TR = 2,250 ms, TE = 3.06 ms, TI = 900 ms, flip angle = 9 deg, FOV = 256 × 256 mm, voxel size = 1.0 × 1.0 × 1.0 mm, number of slices = 208).

**fMRI preprocessing and data sample construction.** MRI preprocessing followed an in-house pipeline based on *fMRIprep*, as described previously [55]. For machine learning analyses, data samples were constructed by pairing segments of sound stimuli with corresponding fMRI activity recorded from auditory cortical regions. Each 8-s sound stimulus was divided into three overlapping 4-s windows to augment the dataset. For each window, an fMRI sample was generated by averaging three consecutive functional volumes acquired 2–8 s after the onset of the window. This yielded three fMRI samples per 8-s stimulus presentation.

In the training set, this procedure yielded a total of 14,400 data samples, generated from 1,200 unique stimuli, each repeated four times (1,200 stimuli × 4 repetitions × 3 samples = 14,400 samples). In both the test and attention datasets, each stimulus was presented eight times. Data samples were created either from individual trials or by averaging the fMRI responses across all eight repetitions to enhance the SNR. The main analyses were conducted using these trial-averaged samples, resulting in 150 test samples (50 stimuli × 3 samples = 150 samples) and 144 attention samples (24 stimuli × 2 attention conditions × 3 samples = 144 samples).

**Regions of interest.** ROIs in the AC were defined using a multi-modal cortical parcellation from the HCP [27]. Thirteen anatomical regions within the AC were identified, including A1, LBelt, MBelt, PBelt, RI, A4, A5, TA2, STGa, STSd anterior, STSv anterior, STSd posterior, and STSv posterior in both hemispheres. The combined voxels from these regions were designated as the AC.

## Sound data for model training

Model training was performed using the VGGsound dataset [54], a public repository of audio and video data extracted from YouTube. The dataset contains 190,055 videos with reliably annotated labels across 309 categories. The dataset was split into 156,487 clips for training, 19,056 for validation, and 14,512 for testing, with the validation and test sets balanced across categories. All audio clips were center-cropped to a duration of 4 s. For preprocessing, audio clips were resampled at 22,050 Hz, and log-spectrograms were generated using a short-term Fourier transform with 1,024 bins, 256 hop lengths, and 80 Mel band scales with zero padding, resulting in 345 time frames. The 80 Mel band scales were centered on frequencies ranging from 125 to 7,600 Hz. To ensure compatibility with temporal downsampling during training, the spectrograms ($n_{Mel\ band} \times n_{time\ frame} = 80 \times 345$) were center-cropped along the time axis to a final size of 80 × 336.

## Model components

Multiple DNN models from previous research by Iashin and Rahtu [28] were used for the proposed reconstruction pipeline. Pretrained models and scripts are available at: https://iashin.ai/SpecVQGAN. Specifically, the pre-trained models for VGGish-ish and spectrogram vocoder [56] were used, as they operate independently of sound length. For generating 4-s sounds in alignment with the fMRI experiments, the provided scripts were used to train models for SpecVQGAN and the audio transformer.

**VGGish-ish classifier.** VGGish-ish, a CNN with 13 convolution layers and three fully connected layers, was trained for natural sound recognition. It extracted DNN features from spectrograms, computing unit responses from each layer with dimensions of $n_{spectral} \times n_{channel} \times n_{temporal}$. These extracted DNN features were then reshaped to the format $n_{spectral * channel} \times n_{temporal}$, preserving the temporal dimension for use as conditioning input to the audio transformer model. Six representative layers were selected based on superior decoding performance in a pilot study: Conv1_1 with dimensions of $5,120 \times 336$, Conv2_1 with dimensions of $5,120 \times 168$, Conv3_1 with dimensions of $5,120 \times 84$, Conv4_1 with dimensions of $5,120 \times 42$, Conv5_3 with dimensions of $2,560 \times 21$, and FC3 with dimensions of 309.

**SpecVQGAN.** SpecVQGAN is a variant of the Vector Quantized Variational Autoencoder (VQVAE) model that applies vector quantization methods [57–59] to convert spectrograms into discrete codebook representations. The model consists of an encoder, a vector quantization process, and a decoder.

The codebook encoder is a 2D convolutional stack with skip connections and normalization layers that processes spectrograms into a latent representation. The latent representations are then discretized using a learned codebook dictionary. The codebook decoder mirrors the architecture of the encoder, except for an upsampling layer that doubles the spatial resolution before applying a convolutional kernel with nearest-neighbor interpolation. Following the default parameter settings from the referenced study, the codebook dimension was set to 256. While the original study [28] used 1,024 codes, applying this to the 4-s cropped dataset caused index collapse, limiting the utilization to about 187 codes. To prevent this issue, the number of codes was restricted to 256. As a result, a 4-s spectrogram input of size $80 \times 336$ was compressed by a factor of 16 in both dimensions, resulting in a codebook representation of size $5 \times 21 (n_{spectral} \times n_{temporal})$.

**Audio transformer.** The audio transformer converted compressed DNN features into codebook representations by predicting codebook indices along the spectral dimension at each temporal point using an autoregressive approach. The model is based on GPT-2-medium, a transformer with 24 layers, 1,024 hidden units, and 16 attention heads. It takes as input DNN features extracted by VGGish-ish, which are reshaped into a sequence of shape $n_{spectral * channel} \times n_{temporal}$. The transformer processes the input sequence in an autoregressive manner. At each temporal point, the transformer generates a probability distribution for the next codebook index using a 256-way softmax classifier. The model was trained to minimize cross-entropy loss between predicted and actual codebook representations, converting the sequence of DNN features into a sequence of codebook representations with dimensions of $5 \times 21 (n_{spectral} \times n_{temporal})$.

For sound reconstruction from different DNN layers, a separate audio transformer was trained for each layer. Since the model was designed to process input with $n_{temporal} = 21$, DNN features from layers with finer temporal resolutions (lower than Conv5) required temporal downsampling. The temporal sampling method used in the referenced study [28] could lead to significant information loss when downsampling high-resolution DNN features. To mitigate this issue, an averaging approach was used, dividing the temporal sequence into 21 segments and averaging the DNN features within each segment. For FC3, identical values were used across the 21 temporal points. As a result, the input to the audio transformer for each DNN layer consisted of DNN features with a shape of $n_{spectral * channel} \times n_{temporal}$, corresponding to Conv1_1 with dimensions of $5,120 \times 21$, Conv2_1 with dimensions of $5,120 \times 21$, Conv3_1 with dimensions of $5,120 \times 21$, Conv4_1 with dimensions of $5,120 \times 21$, Conv5_3 with dimensions of $2,560 \times 21$, and FC3 with dimensions of $309 \times 21$, respectively.

### Target features for decoding

**DNN features.** The unit responses from the highest convolutional layer, Conv5_3, of the VGGish-ish model served as the primary decoding target. The extracted DNN features had dimensions of $2560 \times 21 (n_{spectral * channel} \times n_{temporal})$.

**Spectrogram pixel features.** The pixel values of the spectrogram were computed from all stimuli presented in the fMRI experiments, with each stimulus producing a spectrogram with dimensions of $80 \times 336 (n_{Mel\ band} \times n_{time\ frame})$.

**Spectrotemporal modulation features.** Spectrotemporal modulation feature extraction followed the methods outlined in [22]. First, spectrograms were computed using a filterbank of 128 overlapping bandpass filters spaced logarithmically between 180 and 7,040 Hz. The output was then further processed through bandpass filtering, frequency-axis differentiation, half-wave rectification, and short-term temporal integration. Modulation content was extracted by applying a bank of 2D filters that decompose the spectrogram into spectral and temporal modulation components. These filters were tuned to 6 spectral modulation rates (ranging from 0.5 to 4 cycles per octave) and 10 temporal modulation rates (ranging from 1 to 30 Hz). For each temporal rate, two filters were used to separately capture upward and downward directions, resulting in a total of 20 temporal modulation filters. This process yielded modulation feature with dimensions of $128 \times 40 \times 6 \times 20 (n_{frequency\ channel} \times n_{time\ frame} \times n_{spectral\ modulation\ rate} \times n_{temporal\ modulation\ rate})$. The spectrograms used for extracting modulation features differed from those used in decoding and reconstruction, as it was optimized based on the referenced study.

## Feature decoding analysis

Following the brain decoding framework used in previous vision studies [55,60,61], feature decoders were trained using 14,400 samples from the training dataset for each auditory feature and brain region. The training process began with voxel selection based on the correlation coefficient with the target features using the training stimuli. A total of 500 voxels from AC and 200 from each individual ROI were selected as input to the decoders. The responses of these selected voxels were normalized using the mean and standard deviation computed from the training samples. Like-wise, target auditory features were normalized using the mean and standard deviation computed from the training set. For decoding, we used an L2-regularized linear regression model to predict normalized feature values from the normalized multi-voxel fMRI patterns. The regularization parameter was fixed at 100 for all models, regardless of ROI, feature type, layer, or subject.

In the testing phase, each fMRI sample from the test dataset was normalized using the mean and standard deviation obtained from the training dataset. The trained decoders were applied to these normalized samples to predict auditory features from 150 test samples. The decoded features were then denormalized using the mean and standard deviation for each feature derived from the training dataset. We performed trial-averaging for the test fMRI data, while each trial was treated as an independent sample for the training fMRI data. To mitigate potential discrepancies in the decoded feature distributions due to averaging, we applied a post hoc normalization, scaling the decoded feature values by a factor of $\sqrt{n}$, where n is the number of trials averaged.

Decoding performance was assessed using two metrics: (1) the Pearson correlation coefficient between actual and decoded auditory features across test stimuli in each pixel or unit, and (2) an identification analysis evaluating the ability of decoded auditory features to identify actual stimuli from a set of candidate stimuli. Although the primary results presented in this paper utilize Pearson correlation, we verified that similar results were observed when applying Spearman correlation as an alternative measure.

For the attention experiment, the feature decoders trained under passive listening conditions were used to predict auditory features from 144 fMRI samples in the attention dataset. The decoded auditory feature values were used as inputs to the reconstruction pipeline. Decoding performance was assessed through an identification analysis comparing the correlations of decoded auditory features with the attended and unattended stimuli.

## Reconstruction pipelines

**Full pipeline.** The reconstruction pipeline integrates the trained model components and the feature decoder in a sequential manner. First, the feature decoder extracts DNN features from fMRI responses in the test dataset. These decoded DNN features are then transformed into codebook representations using an audio transformer. The codebook

representations are subsequently converted into spectrograms via a codebook decoder. Finally, a spectrogram vocoder synthesizes audio waveforms from the spectrograms.

**Brain-to-codebook pipeline.** The pipeline begins with training a decoder to predict codebook representations directly from fMRI responses, generating decoded codebook representations from the test dataset. These decoded representations are then transformed into quantized codebook representations using a pre-trained codebook dictionary. The quantized codebook representations are subsequently converted into spectrograms using the codebook decoder. In the final stage, the spectrograms are transformed into audio waveforms using the spectrogram model.

**Pixel optimization pipeline.** An image feature-based optimization technique [62] was refined to optimize pixel values in 2D spectrogram images using the VGGish-ish model. The implementation is based on open-source code available at https://github.com/KamitaniLab/DeepImageReconstruction. The algorithm initializes with a noisy image and iteratively optimizes pixel values to align DNN features extracted by the VGGish-ish model with those decoded from brain activity across all DNN layers. Unlike methods designed for RGB images, this approach was tailored for grayscale spectrograms. A challenge similar to that reported in the referenced study was raised: Loss convergence failed when using only a single layer, specifically a higher layer. To address this, loss minimization was applied across all VGGish-ish layers in comparison with the decoded features. All other parameters remained at their default settings.

## Evaluation of reconstructed sounds

**Identification by human raters.** To evaluate the perceptual similarity between reconstructed sounds and their original stimuli, we conducted a behavioral rating experiment. Human raters listened to a reconstructed sound from an fMRI sample (corresponding to a 4-s segment of an 8-s stimulus) and selected which of two candidates—the original sound or a lure from the test dataset—more closely resembled the reconstruction. Each reconstructed sound was compared against 5 lure candidates, which included two from the Speech category and one from each of the remaining categories. In the attention test, raters were presented with a reconstructed sound and asked to judge whether it more closely matched the attended or unattended stimulus from the same trial. For practical reasons, the experiment was limited to a single fMRI sample per test stimulus and focused on reconstructions from the AC. A total of 17 raters participated in the study, each evaluating approximately 160 unique stimulus pairs. Identification accuracy was defined as the proportion of samples in which the original stimulus was correctly identified. For each sample, this resulted in a score ranging from 0/5–5/5. In the attention test, identification for each sample was scored as a binary outcome based on whether the attended or unattended stimulus was judged to be more similar.

**Identification by extracted feature similarity. Extracted features.** Reconstructed sounds were evaluated based on multiple levels of acoustic representation. At the most basic level, spectrogram pixel values were used as raw features to directly compare the time-frequency structure of reconstructed sounds with their original stimuli. To capture hierarchical representations, we employed the Melception classifier, a DNN trained for sound recognition tasks [28]. Features were extracted from six representative layers of Melception, spanning from early convolutional stages to fully connected layers (Stage 1: Conv1, Stage 2: Conv5, Stage 3: Mix5_d, Stage 4: Mix6_d, Stage 5: Mix7_c, and Stage 6: FC1), thereby enabling assessment across progressively deeper levels of auditory abstraction.

In addition to these features, we also assessed the quality of reconstructed sounds using three standard acoustic attributes. Fundamental frequency (F0) was estimated using the PYIN algorithm [63], SC was computed with the Librosa toolbox (https://librosa.org), and HNR was derived using a publicly available Python implementation (https://github.com/brookemosby/Speech_Analysis). When F0 or HNR could not be computed due to the absence of a clear harmonic structure or a discernible pitch, the corresponding stimuli were excluded from analysis.

For each stimulus and its reconstruction, the mean value was used to represent F0 and HNR, while the median was used for SC, reflecting the typical central tendency of each acoustic feature.

**Identification procedure.** The identification procedure based on feature similarity followed the same structure as the behavioral identification task with human raters, except that the similarity judgment was computed automatically using feature comparisons rather than subjective ratings. For each reconstructed sound, similarity was evaluated between its extracted features and those of two candidate stimuli. Unlike the behavioral evaluation, which used 5 lures drawn from specific categories, this procedure used all 49 other test sounds as potential lures, generating 49 pairwise comparisons per reconstructed sound. Similarity was quantified using Pearson correlation for high-dimensional features, such as spectrogram pixels and DNN representations. For the acoustic features, including the mean F0, median SC, and mean HNR, Euclidean distance was used instead. A pair was considered correctly identified when the reconstructed sound showed greater similarity to the original stimulus than to the lure. The identification accuracy for a sample was defined as the proportion of correctly identified pairs out of the 49 comparisons. In the attention test, the same procedure was applied, but the candidates were limited to the attended and unattended stimuli, and identification was considered correct when the reconstructed sound was more similar to the attended stimulus.

## Identification using disrupted sounds

**Textured sounds.** Sound textures were computed using the Matlab toolbox (http://mcdermottlab.mit.edu/Sound_Texture_Synthesis_Toolbox_v1.7.zip) with default parameters that replicated the synthesis results from McDermott and Simoncelli [64]. The identification task compared the extracted features of a reconstructed sound with those of the textured true sound and those of an original lure candidate, with 49 evaluations conducted for each reconstructed sound.

**Shuffled sounds.** To introduce temporal perturbations, spectrograms of the original stimuli were divided into equal-sized time windows and the resulting segments were randomly shuffled. This manipulation disrupted the temporal structure while preserving spectral content within each segment. Nine different window sizes (12, 24, 48, 96, 190, 286, 500, 1,000, and 2,000 ms) were used to vary the degree of temporal scrambling. After shuffling, the spectrograms were checked and adjusted to ensure they differed from the original, preventing accidental preservation of temporal order. In the identification task, each reconstructed sound was compared against two candidates: the temporally shuffled version of its corresponding original stimulus and the unshuffled version of a lure stimulus. Feature similarity was computed between the reconstruction and each of the two candidates, and identification accuracy was assessed across all 49 test stimuli for each reconstructed sound at each window size.

## Statistics

All statistical analyses were conducted at the individual-subject level, treating data from each subject as an independent replication of the experiment [65]. To assess whether the mean identification accuracy across test stimuli exceeded the chance level of 50%, a 95% confidence interval was used. For the natural sound identification experiment, the sample size was fixed at 50 reconstructed sounds prior to data collection. This number exceeds the sample size required to detect a medium effect size (Cohen's $d = 0.5$) at a significance level of 0.05, which would require $N = 27$. In the attention experiment, performance was evaluated using a binomial test to determine whether the proportion of correctly identified trials was significantly above the 50% chance level. The number of attention samples was fixed at 48 per subject, which is greater than the minimum required to detect a small-to-medium effect size ($g = 0.2$, corresponding to a correct rate of 0.7) with statistical significance at $p < 0.05$, where $N = 37$.

Although each decoder was trained and applied to 4-s samples, statistical analyses were based on 8-s stimulus blocks to address the non-independence of the three decoded samples originating from the same stimulus block. This procedure resulted in 50 data points per subject. For the single sound presentation test, the identification accuracies of the three 4-s samples derived from the same 8-s block were averaged, yielding a single data point per 8-s stimulus. This procedure resulted in 50 data points per subject. For the attention test, the three binary identification outcomes from the same 8-s

block were aggregated using a majority-vote rule to produce a single binary response, resulting in 48 data points per subject. In post-hoc leave-category-out analyses, the number of available data points was reduced relative to the full design. Specifically, there were 10 data points per subject for the animal, music, and environmental sound categories, and 20 data points for the speech category.

## Supporting information

**S1 Fig. Codebook representation. (A)** Training. The SpecVQGAN (Spectrogram Vector Quantized Generative Adversarial Network) model, inspired by the Vector Quantized Variational Autoencoder (VQVAE) framework, is trained to encode spectrograms into compact representations and reconstruct them efficiently. The process begins with an input spectrogram that is transformed into latent representations by the codebook encoder. These representations are quantized into discrete codebook indices using a codebook dictionary, reducing the spectrogram's dimensionality by a factor of 16. For example, a 4-s spectrogram with dimensions of ($80 \times 336$) can be efficiently reduced to a ($5 \times 21$) grid of codebook indices, though it is depicted as a ($2 \times 3$) grid for simplified visual representation. The codebook indices are decoded back into quantized latent representations using the same dictionary and reconstructed into spectrograms by the codebook decoder. The model is trained using the VGGSound dataset, optimizing perceptual, adversarial, codebook, and reconstruction losses. **(B)** Histogram of codebook indices. A histogram displays the frequency distribution of codebook indices for the experimental stimulus dataset, generated using the SpecVQGAN model trained on the VGGSound dataset. The histogram showed a distributed use of codebook indices and maintained relative balance. **(C)** Interpretation of codes as spectrogram patches. To visualize the spectral and temporal patterns captured by individual codes, a set of indices consisting of identical codes is input into the codebook decoder, and the averaged segments highlight the characteristic patterns represented by each code. **(D)** Examples of SpecVQGAN codes. Among the 256 trained codes, 40 examples are illustrated, demonstrating the model's capability to represent and reconstruct complex spectrograms. These findings highlight the ability of SpecVQGAN codes to capture intricate spectral and temporal features through various combinations of discrete codes.
(TIFF)

**S2 Fig. Behavioral evaluation of reconstructed sound. (A)** Stimulus presentation phase. During the behavioral evaluation, participants were sequentially presented with three sounds: the reconstructed sound (target) and two candidate sounds—one corresponding to the true stimulus and the other a false candidate randomly selected from the test set. The currently playing sound was visually highlighted in red. Participants were instructed to assess which of the two candidate sounds was more similar to the target sound. **(B)** Selection phase. After listening to all three sounds, participants selected the sound they judged to be more similar to the target by clicking on the white circle corresponding to either Candidate 1 or Candidate 2.
(TIFF)

**S3 Fig. Reconstruction using different auditory features and their conventional pipelines. (A)** Reconstructed spectrograms using various auditory features with conventional pipelines (Regions of interest (ROI): Auditory cortex (AC); for reconstructed sounds, see https://www.youtube.com/watch?v=KXcDOTOP0h8). The first row shows the original spectrogram of the presented sound. Rows two to four display reconstructions based on different auditory features decoded from AC. The spectrogram-based reconstruction method involves training a model to predict each pixel value of the spectrogram from fMRI responses, followed by conversion into sound waves using a spectrogram vocoder. The modulation feature-based reconstruction employs a two-step process: modulation features were transformed into a spectrogram and then converted into sound waves following a previous study [21]. Reconstructions derived from spectrogram features appear as temporally smooth versions of the original spectrograms. However, reconstructions from modulation features fail to capture distinct spectrotemporal patterns across different frequency ranges, resulting in diminished differentiation between stimuli. **(B)** Evaluation of reconstructed sounds. Each panel corresponds to a specific reconstruction pipeline.

Each bar indicates mean identification accuracy for each subject, with error bars showing the 95% confidence interval (CI) based on 50 data points. Quantitative evaluations reveal that reconstructions using deep neural network (DNN) features outperform those based on other auditory features across most metrics. Spectrogram-based reconstructions achieve an identification accuracy of approximately 70% for pixel-level and lower hierarchical representations, with accuracy declining at higher hierarchical stages. Reconstructions based on modulation features identify stimuli at almost chance levels and showed inferior performance compared to DNN-based reconstructions. The data underlying this figure are provided in S2 Data.
(TIFF)

**S4 Fig. Reconstruction using different auditory features and the audio generator. (A)** Reconstructed spectrograms using various auditory features (ROI: AC; for reconstructed sounds, see https://www.youtube.com/watch?v=5ZH4upF-gi0l). The first row presents the original spectrogram of the presented sound. Rows two to four display reconstructions generated from different auditory features decoded from AC. To examine the influence of different auditory features on reconstruction, only the input (DNN features) is replaced with spectrogram and modulation features while keeping other pipeline components unchanged. The audio generator is trained separately for each auditory feature. Notably, due to YouTube's data availability, the modulation feature-based audio generator was trained on a dataset of 132,551 clips, whereas the DNN and spectrogram feature-based generators used 156,487 clips. Each decoded auditory feature is then used as input to its corresponding audio generator for reconstruction. Reconstructions derived from spectrogram and modulation features exhibit temporally smoothed patterns and partially captured spectrotemporal patterns across different frequency ranges. **(B)** Evaluation of reconstructed sounds. Each panel corresponds to a specific reconstruction pipeline. Bars indicate the mean identification accuracy for each subject, with error bars representing the 95% CI based on 50 data points. Spectrogram-based reconstructions achieve approximately 70% identification accuracy across most evaluation metrics. Reconstructions based on modulation features outperform those using spectrograms but showed lower performance compared to the proposed method using DNN features. Specifically, reconstructions based on modulation features exhibit inferior performance in higher hierarchical stages compared to DNN-based reconstructions. These results indicate that using DNN features for reconstruction outperforms other auditory feature-based approaches, even when utilizing the same reconstruction pipeline. The data underlying this figure are provided in S2 Data.
(TIFF)

**S5 Fig. Effect of model components. (A)** Overview of reconstruction pipelines. The brain-to-codebook pipeline predicts codebook representations directly from fMRI responses (brown line), which are then transformed into spectrograms and sound waves via the codebook encoder and decoder. The pixel optimization pipeline iteratively adjusts spectrogram pixel values to match decoded DNN features inferred from fMRI responses (orange line). **(B)** Reconstructed spectrograms from different pipelines (ROI: AC; for reconstructed sounds, see https://www.youtube.com/watch?v=DFLsaLY4g64). The first row displays the original spectrograms of the stimuli, and subsequent rows show reconstructions from each pipeline. The brain-to-codebook pipeline produces temporally smoothed spectrograms, while the pixel optimization pipeline generates spectral patterns with specific frequency alignment but noticeable noise and limited temporal fidelity, resulting in texturally noisy outputs. **(C)** Evaluation of reconstructed sounds. Each panel represents mean identification accuracy per subject for a given reconstruction pipeline, with error bars denoting the 95% CI based on 50 data points. The brain-to-codebook pipeline achieves approximately 65% identification accuracy based on spectrogram pixels but showed declining performance at higher hierarchical stage. The pixel optimization pipeline performs better than the brain-to-codebook approach in both spectrogram pixel accuracy and higher hierarchical stage but still fell short of the proposed model. These results highlight that while both pipelines could reconstruct spectral patterns, only the proposed model consistently preserved perceptual qualities similar to the original stimuli. The data underlying this figure are provided in S2 Data.
(TIFF)

**S6 Fig. Evaluation of reconstructed sounds using candidate stimuli from the same category.** Identification accuracy for reconstructed sounds is presented by category. Each bar represents the mean accuracy across 10 test stimuli for the animal, music and environment categories and 20 test stimuli for the speech category. Error bars indicate the 95% CI, and each color corresponds to an individual subject. Human ratings are generally above 70% for animals and environments, while speech ratings ranged from 60% to 80%, with music showing greater variability. Trends in human ratings closely align with objective evaluations. These results suggest that reconstructed sounds for animals and environments reliably captured category-specific features, enabling accurate identification of true stimuli, whereas reconstructions for speech and music lacked sufficient detail for consistent within-category discrimination. The data underlying this figure are provided in S2 Data. (TIFF)

**S7 Fig. Single-trial reconstruction. (A)** Reconstructed spectrograms (ROI: AC, DNN layer: Conv5; for reconstructed sounds, see https://www.youtube.com/watch?v=exUeKzT0Qfo). The top row shows the original spectrogram of the stimulus sounds. The second row depicts reconstructions based on fMRI samples averaged over eight trials. Rows three to six display reconstructions using single-trial fMRI samples from different trials. **(B)** Evaluation of reconstructed sounds. Each panel represents the mean identification accuracy for each subject. The dark blue bar shows results from fMRI samples averaged over eight trials, while the orange bar indicates the average accuracy of single-trial reconstructions across eight repetitions. Error bars represent the 95% CI based on 50 data points. Identification accuracy for single-trial reconstructions is approximately 60% for spectrogram and lower hierarchical stages, improving to around 80% in higher hierarchical stages. For acoustic features, the model achieves 60% accuracy for F0 and SC. Despite lower performance compared to trial-averaged reconstructions, these findings confirm the model's capability to produce reasonable reconstructions from single-trial fMRI samples. The data underlying this figure are provided in S2 Data. (TIFF)

**S8 Fig. Evaluation of reconstructed sound using textured true stimuli for all subjects.** Identification accuracy of reconstructed sounds is shown for each subject when compared against temporally textured true stimuli. Each panel corresponds to an individual subject. Dark blue bars represent accuracy using the original true stimuli, and orange bars indicate accuracy using the textured true stimuli. Error bars denote the 95% CIs based on 50 data points. The data underlying this figure are provided in S2 Data. (TIFF)

**S9 Fig. Evaluation of reconstructed sound using temporally shuffled true stimuli for all subjects.** The identification accuracy of reconstructed sounds was assessed using temporally shuffled true stimuli. Each panel corresponds to an individual subject, with each bar representing the mean accuracy for different segment sizes, indicated by varying colors. Error bars denote the 95% CIs based on 50 data points. The data underlying this figure are provided in S2 Data. (TIFF)

**S10 Fig. Decoding performance of auditory areas and DNN layers. (A)** Identification accuracy across auditory areas. Identification accuracy for decoded DNN features from the Conv5 layer is presented for different auditory regions. Each bar represents the average accuracy for individual subjects, with error bars indicating the 95% CI based on 50 data points. Colors correspond to individual subjects. Accuracy consistently exceeds 70% across primary and secondary auditory areas, with minimal variation among regions. These results suggest a distributed auditory processing system rather than a strict hierarchical correspondence between auditory regions. **(B)** Identification accuracy across DNN layers. Identification accuracy for decoded features from various DNN layers using the AC is shown. Performance improves at higher DNN layers compared to lower layers, demonstrating that hierarchical representations encoded in different DNN layers can be reliably predicted from the AC. The data underlying this figure are provided in S2 Data. (TIFF)

 

**S11 Fig. Reconstruction from different auditory areas and DNN layers. (A)** Reconstructed spectrograms from individual ROIs (DNN layer: Conv5; for reconstructed sounds, see https://www.youtube.com/watch?v=qjMg887OOTM). The top row shows the original spectrogram of the presented sound. Rows two through seven display spectrograms reconstructed using fMRI data from individual ROIs, highlighting the contributions of each auditory region. **(B)** Reconstructed spectrograms using different DNN feature layers (ROI: AC; for reconstructed sounds, see https://www.youtube.com/watch?v=nz-R0ibTYmTI). The top row shows the original spectrogram of the presented sound. Rows two through seven represent spectrograms reconstructed using features from different layers of the DNN model, illustrating the influence of hierarchical DNN features on reconstruction quality.
(TIFF)

**S12 Fig. Effect of fully connected layers. (A)** Reconstructed spectrograms generated using decoded features from three DNN layers: the last convolutional layer (Conv5) and two fully connected layers (FC1 and FC3) of the VGGish-ish model. The top row shows the original spectrograms of the stimulus sounds. The following rows display spectrograms reconstructed using features decoded from each layer. **(B)** Evaluation of reconstructed sounds from Conv5, FC1, and FC3 (ROI: AC). Each bar represents the mean identification accuracy calculated for each subject, with the error bar indicating the 95% CI estimated from 50 data points. While Conv5 tended to show slightly higher performance in pixel-based measures, the overall differences among the three layers were modest and varied across subjects. These results suggest that both convolutional and fully connected layers retain sufficient information for reconstructing perceptual attributes. The data underlying this figure are provided in S2 Data.
(TIFF)

**S13 Fig. Recovery check of DNN layers. (A)** Recovered spectrograms from different DNN layers. The top row shows the original spectrograms of the stimulus sounds, while rows two through seven display spectrograms recovered using true DNN features from different layers. The first and second columns show natural sounds selected from VGGSound, while the third to fifth columns present artificial sounds not included in model training. Recovery from lower DNN layers (Conv1 to Conv3) achieves near-perfect reconstruction of the original spectrograms. Spectrograms from Conv4 and Conv5 show subtle degradation but retain key spectral patterns. Recovery using category-level features from FC3 exhibits more pronounced degradation, particularly in temporal structure, although spectral patterns often remained discernible. Even for artificial sounds, recovery demonstrates high fidelity. **(B)** Evaluation of recovered sounds from different DNN layers. Bars represent the mean identification accuracy, with error bars indicating the 95% CI based on 50 data points. Recovery from lower layers achieves nearly 100% accuracy. For Conv5, accuracy remains close to 95% in pixel-based evaluations, with a slight decline in lower hierarchical stages and near-perfect recovery (~100%) in higher hierarchical stages. Recovery from FC3 shows significant degradation in pixel-based and lower hierarchical stages but retained over 90% accuracy for higher hierarchical stages. The data underlying this figure are provided in S2 Data.
(TIFF)

**S14 Fig. Reconstruction using untrained DNN features. (A)** Evaluation of decoded features across DNN layers extracted from an untrained VGGish-ish model (ROI: AC). Decoded features from the trained model show lower performance than those from the untrained model, particularly in Conv5 (our primary decoding target). At lower hierarchical layers, such as Conv1 and Conv2, there are minimal difference between trained and untrained features. However, as layer depth increases, trained DNN features exhibit a gradual performance improvement, while untrained features remain relatively unchanged. **(B)** Reconstructed spectrograms using different DNN feature layers from an untrained model (see https://www.youtube.com/watch?v=nX6idzcQGMU). The top row presents the original spectrogram of the presented sound, while rows two through seven display reconstructions based on features from different layers of the untrained DNN model. Although these reconstructions reflect the influence of hierarchical DNN features on reconstruction quality, they poorly resemble the original spectrograms. However, reconstructions from the FC3 layer retain some distinct

spectrotemporal patterns. **(C)** Evaluation of reconstructed sounds. The difference between reconstructions using trained and untrained DNN features is marginal. Beyond this, however, a pattern of hierarchical correspondence is observed. At higher hierarchical stages, reconstructions from lower layers show lower performance when using trained DNN features compared to untrained. However, for higher layers, trained DNN features outperform untrained features, showing a shift in performance across layers. These results suggest that while model architecture influenced reconstruction quality, task-optimized DNN features played a role in capturing hierarchical structure and improving perceptual evaluations. The data underlying this figure are provided in S2 Data.
(TIFF)

**S15 Fig. Evaluation of reconstructed sounds using hierarchical representation with untrained filter weights.** Reconstructed sounds (ROI: AC) were evaluated using a trained Melception model (left panel) and an untrained model with random filter weights (right panel). Each bar represents the mean identification accuracy per subject, with error bars indicating the 95% CI based on 50 data points. Lower hierarchical stages from the untrained model show minimal changes compared to the trained model. However, performance declines across all DNN layers as the representation hierarchy increases. Reconstructions from Conv5 exhibit a nearly 30% drop in performance at the hierarchical stage 4, which aligns most closely with human ratings. These findings underscore the critical role of task-specific optimization in evaluating hierarchical representations and suggest that the observed hierarchical structure in reconstructed sounds is not solely attributed to the model's architectural characteristics. The data underlying this figure are provided in S2 Data.
(TIFF)

**S16 Fig. Sound reconstruction with attention from different auditory areas. (A)** Reconstructed spectrograms from individual ROIs during selective auditory attention tasks (DNN layer: Conv5; for reconstructed sounds, see https://www.youtube.com/watch?v=jJDBq2Ix0Aw). The top panel shows the spectrograms of two superimposed sounds presented during the task, where subjects were instructed to focus on one specific sound. The bottom panel displays spectrograms reconstructed from each individual ROI for Subject S4. **(B)** Identification accuracy of attended sound from individual ROIs. Each panel corresponds to a specific auditory area, with each bar representing the mean identification accuracy for an individual subject based on 48 data points. The dashed line indicates the significance threshold ($p < 0.05$) as determined by a binomial test. These findings highlight the ability of individual auditory regions to contribute to distinguish attended sounds in the presence of competing auditory stimuli. The data underlying this figure are provided in S2 Data.
(TIFF)

**S1 Movie. Reconstructed sounds under natural sound listening conditions.** A side-by-side comparison of spectrograms for actual stimuli (left) and reconstructed sounds (right) across various natural sound stimuli. (Subjects: S3, S4, and S5; ROI: AC; DNN layer: Conv5). https://www.youtube.com/watch?v=kNSseidxFJU.
(MP4)

**S2 Movie. Reconstructed sounds using different auditory features with conventional pipelines.** (Subject: S3, ROI: AC) https://www.youtube.com/watch?v=KXcDOTOP0h8.
(MP4)

**S3 Movie. Reconstructed sounds using different auditory features with the same reconstruction pipelines.** (Subject: S3, ROI: AC) https://www.youtube.com/watch?v=5ZH4upFgi0I.
(MP4)

**S4 Movie. Reconstructed sounds using different reconstruction pipelines.** (Subject: S3, ROI: AC) https://www.youtube.com/watch?v=DFLsaLY4g64.
(MP4)

**S5 Movie. Single-trial reconstruction.** (Subject: S3, ROI: AC, DNN layer: Conv5) https://www.youtube.com/watch?v=exUeKzT0Qfo.
(MP4)

**S6 Movie. Reconstructed sounds using brain decoder with ablated training category set.** (Subject: S3, ROI: AC, DNN layer: Conv5) https://www.youtube.com/watch?v=znm6NWL1YYY.
(MP4)

**S7 Movie. Reconstructed sounds using individual auditory regions.** (Subject: S3, DNN layer: Conv5) https://www.youtube.com/watch?v=qjMg887OOTM.
(MP4)

**S8 Movie. Reconstructed sounds using different DNN layers.** (Subject: S3, ROI: AC) https://www.youtube.com/watch?v=nzR0ibTYmTI.
(MP4)

**S9 Movie. Reconstructed sounds using untrained DNN features.** (Subject: S3, ROI: AC) https://www.youtube.com/watch?v=nX6idzcQGMU.
(MP4)

**S10 Movie. Reconstructed sounds under selective auditory attention tasks.** The left panel shows the spectrogram of superimposed stimuli from two distinct sound categories (illustrated with different colors). The right panel displays the spectrogram of reconstructed sounds when the subject focuses attention on one of the two sounds. (Subjects: S4 and S5; ROI: AC; DNN layer: Conv5). https://www.youtube.com/watch?v=1ZHCoiyqPb4.
(MP4)

**S11 Movie. Reconstructed sounds under selective auditory attention tasks using individual auditory regions.** (Subject: S4, DNN layer: Conv5) https://www.youtube.com/watch?v=jDBq2lx0Aw.
(MP4)

**S1 Data. Data used for Fig 3C analysis.** This file contains the dataset used to generate the results presented in Fig 3C.
(json)

**S2 Data. Data used for all figure analyses (excluding Fig 3C).** This file includes all datasets used to generate the main and supporting figures, excluding the data used in Fig 3C, which is provided separately in S1 Data.
(XLSX)

## Acknowledgments

This study was conducted using the fMRI scanner and related facilities of the Kyoto University Institute for the Future of Human Society. The authors thank Eizaburo Doi, Fan Cheng, and Ken Shirakawa for their helpful discussion and comments on the manuscript.

## Author contributions

**Conceptualization:** Jong-Yun Park, Yukiyasu Kamitani.

**Data curation:** Jong-Yun Park, Mitsuaki Tsukamoto, Misato Tanaka, Yukiyasu Kamitani.

**Formal analysis:** Jong-Yun Park, Yukiyasu Kamitani.

**Funding acquisition:** Yukiyasu Kamitani.

**Investigation:** Jong-Yun Park, Yukiyasu Kamitani.

**Methodology:** Jong-Yun Park, Yukiyasu Kamitani.

**Project administration:** Yukiyasu Kamitani.

**Resources:** Yukiyasu Kamitani.

**Software:** Jong-Yun Park, Mitsuaki Tsukamoto, Misato Tanaka.

**Supervision:** Yukiyasu Kamitani.

**Validation:** Yukiyasu Kamitani.

**Visualization:** Jong-Yun Park, Yukiyasu Kamitani.

**Writing – original draft:** Jong-Yun Park, Yukiyasu Kamitani.

**Writing – review & editing:** Jong-Yun Park, Yukiyasu Kamitani.

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
