## [Editor Report · Decision Letter 0]

Dear Dr Park,

Thank you for submitting your manuscript entitled "Sound reconstruction from human brain activity via a generative model with hierarchical auditory features" for consideration as a Research Article by PLOS Biology.

Your manuscript has now been evaluated by the PLOS Biology editorial staff[as well as by an academic editor with relevant expertise and I am writing to let you know that we would like to send your revised manuscript back to the peer reviewer.

Once your full submission is complete, your paper will undergo a series of checks in preparation for peer review. After your manuscript has passed the checks it will be sent out for review. To provide the metadata for your submission, please Login to Editorial Manager (https://www.editorialmanager.com/pbiology) within two working days, i.e. by May 02 2025 11:59PM.

Kind regards,

Christian

Christian Schnell, PhD

Senior Editor

PLOS Biology

cschnell@plos.org

---

## [Decision Letter · Decision Letter 1]

Dear Dr Park,

Thank you for your patience while we considered your revised manuscript "Sound reconstruction from human brain activity via a generative model with hierarchical auditory features" for consideration as a Research Article at PLOS Biology. Your revised study has now been evaluated by the PLOS Biology editors, the Academic Editor and one of the original reviewers.

In light of the reviews, which you will find at the end of this email, we are pleased to offer you the opportunity to address the remaining points from Reviewer 1 in a revision that we anticipate should not take you very long. We will then assess your revised manuscript and your response to the reviewers' comments with our Academic Editor aiming to avoid further rounds of peer-review, although we might need to consult with the reviewers, depending on the nature of the revisions.

**IMPORTANT - SUBMITTING YOUR REVISION**

*Resubmission Checklist*

*Published Peer Review*

*PLOS Data Policy*

*Blot and Gel Data Policy*

Sincerely,

Christian

Christian Schnell, PhD

Senior Editor

PLOS Biology

cschnell@plos.org

REVIEWS:

Reviewer #1: I thank the authors for making a substantial effort to improve the quality of the paper. Overall, the thorough revisions and additions have strengthened the contribution. I especially appreciated the effort taken to run a behavioral study on the generated sounds. For the most part, my concerns have been well addressed, and I found the paper a fascinating read and an impressive engineering feat.

A few remaining things that I believe would help the paper before publication, which can be addressed with fairly minor wording changes:

- I still don't think the claim of lack of hierarchical correspondence for decoding (line 566-568) is appropriately backed up by the presented results. Is this statement just referring to Figure S10A, with the single layer of the DNN decoding each region? Overall, lines 566-576 seem very bold to be making from just this result, and one would have to do substantial control analysis beyond what is currently in the paper to make sure that there are not additional confounds. I encourage the authors to significantly tone down this section.

- It is still not clear how the reconstruction of input provides "crucial insights" into neural coding (first line of the introduction, lines 57-58). This was also brought up in Reviewer 2's concern #8. I appreciate the authors' response to my concern #14; however, the response addresses linear decoding, which is not what the reconstruction process is doing. The main text now seems generally clear what the scientific contribution of the decoder vs. the reconstruction says, but I think some of the abstract and introduction should be modified with this in mind (i.e. first sentence mentioned above).

- The identifiability result with textured and scrambled stimuli at different lengths provides interesting insights into how the reconstructions are performing, however result this is not well captured in the abstract (the line "despite its ability to reconstruct exact temporal sequences" on 52 seems vague). I appreciated the authors interpretation of this result in the discussion and relating it to the nature of the fMRI signal, so perhaps using some similar wording in the abstract would help with clarity.

- For the category specific analysis of the behavioral results (lines 292-294), is the randomly selected alternative always from the same category? Or is this just the average across the same and different categories (I think it is probably this)? As written, it is a little unclear in the main text.

- Also, regarding the behavioral results, can any within-category confusions be reported (i.e. something similar to Fig S6 for the modeling results). For instance, it seems likely that a speech sound is easy to tell apart from thunder, but then the identification of one speech sound from another speech sound is close to chance. This would be good to know and would help the interpretation of the behavioral results.

- I do not think there is enough data to validate lines 326-328. This observation could perhaps be mentioned in the discussion as a future direction, but it seems inappropriate for the results.

- Error bars are not included in Fig 8C. If there is a reason they are not included (for instance if they are too small as mentioned in the rebuttal for other plots), then this should be mentioned in the caption, especially for the human ratings.

---

## [Editor Report · Decision Letter 2]

Dear Dr Park,

Thank you for your patience while we considered your revised manuscript "Sound reconstruction from human brain activity via a generative model with hierarchical auditory features" for publication as a Research Article at PLOS Biology. This revised version of your manuscript has been evaluated by the PLOS Biology editors and the Academic Editor.

Based on our Academic Editor's assessment of your revision, we are likely to accept this manuscript for publication, provided you satisfactorily address the following data and other policy-related requests:

* We would like to suggest a different title to improve its accessibility for our broad audience:

Natural sounds can be reconstructed from human neuroimaging data using a deep neural network

* Please add the links to the funding agencies in the Financial Disclosure statement in the manuscript details.

* Please specify whether the participants provided written or oral consent.

* DATA POLICY:

Regardless of the method selected, please ensure that you provide the individual numerical values that underlie the summary data displayed in the following figure panels as they are essential for readers to assess your analysis and to reproduce it: 3CD, 4DG, 5CF, 6, 7, 8C, S3B, S4B, S5C, S6, S7, S8, S9B, S10, S12B, S13B, S14AC, S15 and S16B.

* CODE POLICY

We expect to receive your revised manuscript within two weeks.

*Published Peer Review History*

*Press*

Sincerely,

Christian

Christian Schnell, PhD

Senior Editor

cschnell@plos.org

PLOS Biology

---

## [Editor Report · Decision Letter 3]

Dear Dr Park,

Thank you for the submission of your revised Research Article "Natural sounds can be reconstructed from human neuroimaging data using deep neural network representation" for publication in PLOS Biology. On behalf of my colleagues and the Academic Editor, Manuel Malmierca, I am pleased to say that we can in principle accept your manuscript for publication, provided you address any remaining formatting and reporting issues. These will be detailed in an email you should receive within 2-3 business days from our colleagues in the journal operations team; no action is required from you until then. Please note that we will not be able to formally accept your manuscript and schedule it for publication until you have completed any requested changes.

PRESS

We frequently collaborate with press offices. If your institution or institutions have a press office, please notify them about your upcoming paper at this point, to enable them to help maximize its impact. If the press office is planning to promote your findings, we would be grateful if they could coordinate with biologypress@plos.org. If you have previously opted in to the early version process, we ask that you notify us immediately of any press plans so that we may opt out on your behalf.

Sincerely, 

Christian

Christian Schnell, PhD

Senior Editor

PLOS Biology

cschnell@plos.org